# Determination of the absorption cross-sections of higher order iodine oxides at 355 nm and 532 nm.

Thomas R. Lewis[1,2], Juan Carlos Gómez Martín[3*], Mark A. Blitz[2], Carlos A. Cuevas[1], John M. C. Plane[2*] and Alfonso Saiz-Lopez[1*]

[1]Department of Atmospheric Chemistry and Climate, Institute of Physical Chemistry Rocasolano, CSIC, Madrid, Spain
[2]School of Chemistry, University of Leeds, LS29JT, Leeds, UK
[3]Instituto de Astrofísica de Andalucía, CSIC, 18008, Granada, Spain

*Correspondence to*: Juan Carlos Gómez Martín (jcgomez@iaa.es), John M. C. Plane (J.M.C.Plane@leeds.ac.uk) and Alfonso Saiz-López (a.saiz@csic.es).

**Abstract.** Iodine oxides ($I_xO_y$) play an important role in the atmospheric chemistry of iodine. They are initiators of new particle formation events in the coastal and polar boundary layer and act as iodine reservoirs in tropospheric ozone-depleting chemical cycles. Despite the importance of the aforementioned processes, the photochemistry of these molecules has not been studied in detail previously. Here, we report the first determination of the absorption cross sections of $I_xO_y$, x = 2, 3, 5, y = 1-12 at $\lambda$ = 355 nm by combining pulsed laser photolysis of $I_2/O_3$ gas mixtures in air with time-resolved photo-ionization time-of-flight mass spectrometry, using $NO_2$ actinometry for signal calibration. The oxides selected for absorption cross section determinations are those presenting the strongest signals in the mass spectra, where signals containing 4 iodine atoms are absent. The method is validated by measuring the absorption cross section of IO at 355 nm, $\sigma_{355\ nm,\ IO} = (1.2 \pm 0.1) \times 10^{-18}$ cm$^2$, which is found to be in good agreement with the most recent literature. The results obtained are: $\sigma_{355\ nm,\ I2O3} < 5 \times 10^{-19}$ cm$^2$ molecule$^{-1}$, $\sigma_{355\ nm,\ I2O4} = (3.9 \pm 1.2) \times 10^{-18}$ cm$^2$ molecule$^{-1}$, $\sigma_{355\ nm,\ I3O6} = (6.1 \pm 1.6) \times 10^{-18}$ cm$^2$ molecule$^{-1}$, $\sigma_{355\ nm,\ I3O7} = (5.3 \pm 1.4) \times 10^{-18}$ cm$^2$ molecule$^{-1}$ and , $\sigma_{355\ nm,\ I5O12} = (9.8 \pm 1.0) \times 10^{-18}$ cm$^2$ molecule$^{-1}$. Photodepletion at $\lambda$ = 532 nm was only observed for OIO, which enabled determining upper limits for the absorption cross sections of $I_xO_y$ at 532 nm using OIO as an actinometer. These measurements are supplemented with ab-initio calculations of electronic spectra in order to estimate atmospheric photolysis rates $J(I_xO_y)$. Our results confirm a high $J(I_xO_y)$ scenario where $I_xO_y$ is efficiently removed during daytime, implying enhanced iodine-driven ozone depletion and hindering iodine particle formation. Possible $I_2O_3$ and $I_2O_4$ photolysis products are discussed, including $IO_3$, which may be a precursor to iodic acid ($HIO_3$) in the presence of $HO_2$.

## 1. Introduction

Photolabile iodine-containing molecules are emitted into the lower atmosphere from the sea surface and from marine biota. The atmospheric processing of iodine leads to its accumulation in aerosol, subsequent transport and deposition on land, where it enters the continental ecosystems (Saiz-Lopez et al., 2012b). In the course of this process, gas-phase reactive iodine is

involved in two important chemical processes of the background troposphere: ozone depletion and new particle formation. Iodine is thought to be responsible for 9 to 16% of the contemporary ozone removal in the tropical troposphere (Saiz-Lopez et al., 2014; Sherwen et al., 2016), and there is evidence that anthropogenic ozone pollution enhances iodine release from the sea surface (Carpenter et al., 2013; Chance et al., 2014; MacDonald et al., 2014; Prados-Roman et al., 2015; Cuevas et al., 2018), which in turn has accelerated ozone loss in the last decades (Cuevas et al., 2018). The atmospheric chemistry of iodine is in principle fairly simple, since the main atmospheric fate of the iodine atoms is reaction with ozone to form iodine monoxide (IO). This radical photolyzes readily (Gómez Martín et al., 2005), creating an ozone-neutral cycle and establishing a steady state concentration of I and IO, which are then termed collectively as active iodine or $IO_x$. Any other chemical cycle involving IO which recycles atomic iodine without concomitant generation of atomic oxygen leads to ozone depletion, e.g. (Gómez Martín et al., 2009):

$$I + O_3 \rightarrow IO + O_2 \hspace{8cm} \text{(R1)}$$

$$IO + IO \rightarrow OIO + I \hspace{7.5cm} \text{(R2)}$$

$$OIO + h\nu \rightarrow I + O_2 \hspace{7.5cm} \text{(R3)}$$

Net:  $2O_3 \rightarrow 3O_2$

Further steps in this scheme lead to the formation of higher iodine oxides (Gómez Martín et al., 2013):

$$IO + IO \rightarrow I_2O_2 \hspace{7.7cm} \text{(R4)}$$

$$IO + OIO \rightarrow I_2O_3 \hspace{7.5cm} \text{(R5)}$$

$$OIO + OIO \rightarrow I_2O_4 \hspace{7.2cm} \text{(R6)}$$

$$I_xO_y + I_xO_y \rightarrow\rightarrow\rightarrow \text{particles} \hspace{5.2cm} \text{(R7)}$$

Previous laboratory experiments on iodine photochemistry have reported kinetic growth of broad band absorptions following the decay of IO and OIO as well as significant deposition of particulates on the walls of the reactors (Cox and Coker, 1983; Laszlo et al., 1995; Harwood et al., 1997; Gómez Martín et al., 2005). These broad band absorbers have been tentatively identified as some of the higher iodine oxides produced in reactions R4, R5 and R6 (Bloss et al., 2001; Gómez Martín et al., 2007), which are believed to be precursors to the particles that eventually deposit on the walls. Furthermore, iodine oxide containing particles have been observed in multiple aerosol flow tube and steady state chamber experiments starting from the photooxidation of iodine precursors directly injected into the chamber (Hoffmann et al., 2001; Jimenez et al., 2003; Saunders et al., 2010; Wei et al., 2017) or emitted from cultures of algae under oxidative stress (McFiggans et al., 2004; Pirjola et al., 2005). Emissions of $I_2$ and alkyl-iodides from coastal macroalgae and polar phytoplankton and concurrent observation of IO and OIO (Saiz-Lopez and Plane, 2004) have been unambiguously linked to intense new particle formation events (Jimenez et al., 2003; McFiggans et al., 2004). Thus, there is a strong indication that iodine oxide clustering also happens in the coastal

boundary layer and is responsible for particle formation events.

Recent Nitrate Ion Chemical Ionization - Atmospheric Pressure Interface- Time of Flight Mass Spectrometry ($NO_3^-$ CI-API-ToF-MS) observations of $IO_3^-$ and $IO_3$-containing ion clusters in coastal and polar environments, as well as complementary laboratory experiments in the absence of $HO_x$, have been interpreted as direct measurements of gas-phase iodic acid ($HOIO_2$, hereafter denoted as $HIO_3$) and $HIO_3$ clusters (Sipilä et al., 2016) by ionization of the ambient species by $NO_3^-$ in the instrument

inlet. Since all possible reaction paths for I, IO and OIO with $H_2O$ are very endothermic (Canneaux et al., 2010; Hammaecher et al., 2011; Khanniche et al., 2017a), and $IO_x$–$H_2O$ complexes are very weakly bound (Galvez et al., 2013), the formation of oxyacids may rather proceed via hydrolysis of higher iodine oxides (Kumar et al., 2018):

$$I_2O_y + H_2O \rightarrow HIO_x + HIO_{y-x+1} \qquad\qquad\qquad\qquad (R8)$$

Ozone depletion and particle formation are to some extent competing processes. Significant $I_xO_y$ photolysis rates may result

in regeneration of $IO_x$:

$$I_2O_2 + h\nu \rightarrow IO + IO \qquad\qquad\qquad\qquad (R9a)$$

$$\rightarrow I + OIO \qquad\qquad\qquad\qquad (R9b)$$

$$I_2O_3 + h\nu \rightarrow IO + OIO \qquad\qquad\qquad\qquad (R10a)$$

$$\rightarrow I + IO_3 \qquad\qquad\qquad\qquad (R10b)$$

$$I_2O_4 + h\nu \rightarrow OIO + OIO \qquad\qquad\qquad\qquad (R11a)$$

$$\rightarrow IO + IO_3 \qquad\qquad\qquad\qquad (R11b)$$

These photochemical reactions would enhance ozone depletion, while slowing down the incorporation of iodine into aerosol via oxides and/or oxyacids, and represent an important source of uncertainty in the iodine chemical mechanism incorporated into global chemistry transport models (Saiz-Lopez et al., 2014; Sherwen et al., 2016). In order to reduce this uncertainty, we

report here a set of $I_xO_y$ photodepletion experiments with mass spectrometric detection, which enables unambiguous observation of all the species of interest (Gómez Martín et al., 2013). Experiments devoted to understanding the interaction between $I_xO_y$ and water, potentially leading to the formation of $HIO_3$, will be reported elsewhere.

## 2. Methods

### 2.1. Experimental set-up

A pulsed laser photolysis (PLP) system has been employed to generate iodine oxides from the photolysis of $O_3$ in the presence of $I_2$ in a tubular reaction cell (Figure 1). A flow of typically 1.5 slm of He carrier gas (99.999%, BOC Gases) was introduced in the reactor. $I_2$ molecules were entrained in the carrier flow by passing a smaller flow of He (~100 sccm) through a 12 mm

diameter temperature-stabilised Teflon tube containing $I_2$ crystals (>99.5%, Sigma-Aldrich). An electrical discharge ozone generator converted ~2% of a ~100 sccm $O_2$ flow (99.999%, BOC Gases) to $O_3$, which was introduced to the main flow via an inlet on the flow tube.

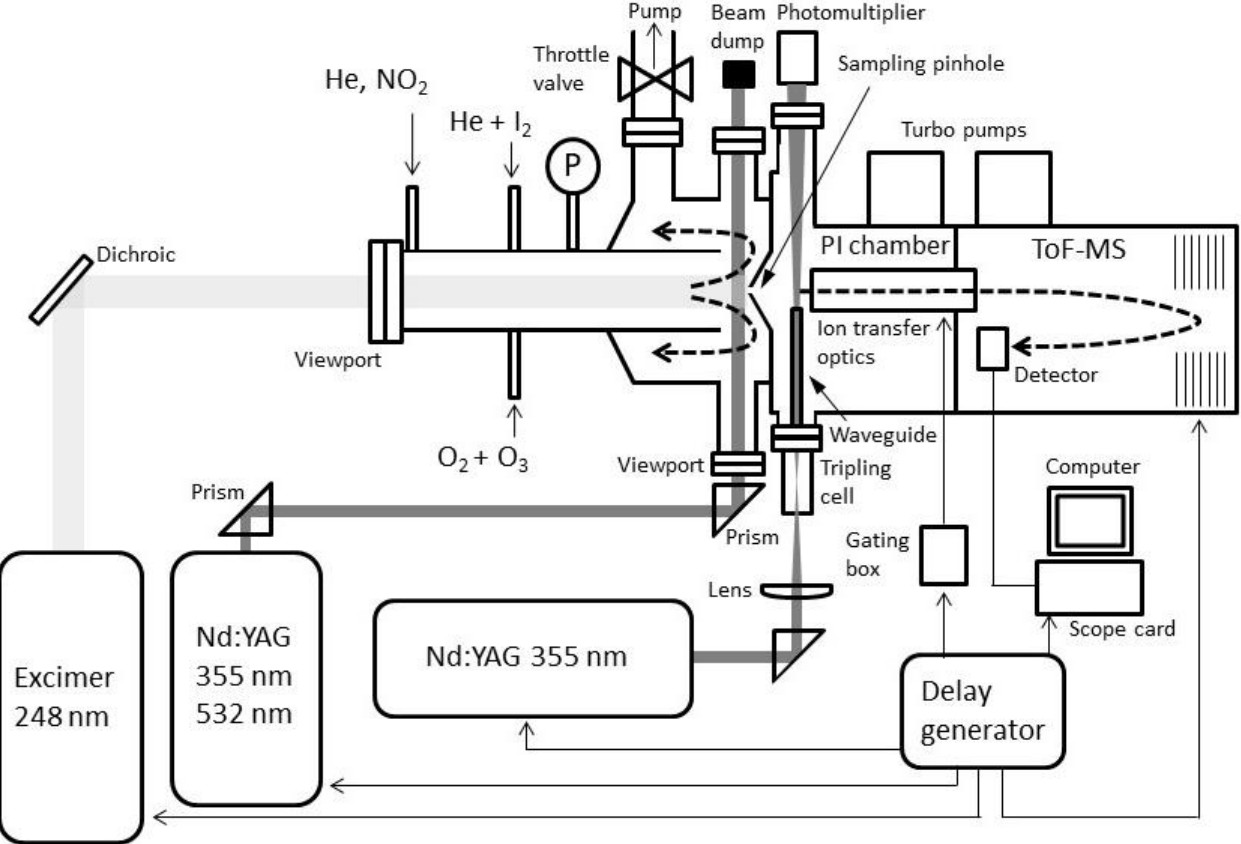

**Figure 1: Schematic diagram of the experimental set-up for studying the photolysis of iodine oxides**

Ozone was photolyzed at 248 nm by an excimer laser beam (Lambda Physik Compex 102), which was passed unfocussed through a quartz viewport along the tube main axis. This generated the well-known sequence of reactions:

$$O_3 + hv \rightarrow O(^1D) + O_2 \tag{R12}$$

$$O(^1D) + M \rightarrow O(^3P) + M \tag{R13}$$

$$O(^3P) + I_2 \rightarrow IO + I \tag{R14}$$

followed by reactions R1, R2, R4, R5 and R6, as well as further $I_xO_y$ clustering reactions. The precursor concentrations ($[I_2]$~ 2-4 × $10^{14}$ molecule cm$^{-3}$, $[O_3]$~ 3-5 × $10^{14}$ molecule cm$^{-3}$) and excimer laser energy (120-190 mJ pulse$^{-1}$, ~50-80 mJ pulse$^{-1}$ cm$^{-2}$) were tuned to ensure that $I_2$ was in excess over $O(^3P)$. Iodine and ozone concentrations were measured using a 532 nm solid state laser in a single pass and a multipass Herriot cell (Lewis et al., 2018), respectively (Figure 2). A reflection (~5%)

of the main 532 nm diode laser (Thorlabs CPS532 532 nm 4.5 mW) beam was directed along the length of the cell (length = 40 cm) and detected using a photodiode detector (Thorlabs SM05PD3A). Ozone was detected by passing the main beam through a hole in the rear of a silver-coated concave mirror (Thorlabs CM508-200-P01 f = 20 cm) onto an identical mirror (without a hole) positioned 40 cm from the first mirror. The light was passed 40 times through the cell, and exits through the same entrance hole (some passes are omitted from the diagram for simplicity), giving a path length of 12 m.`

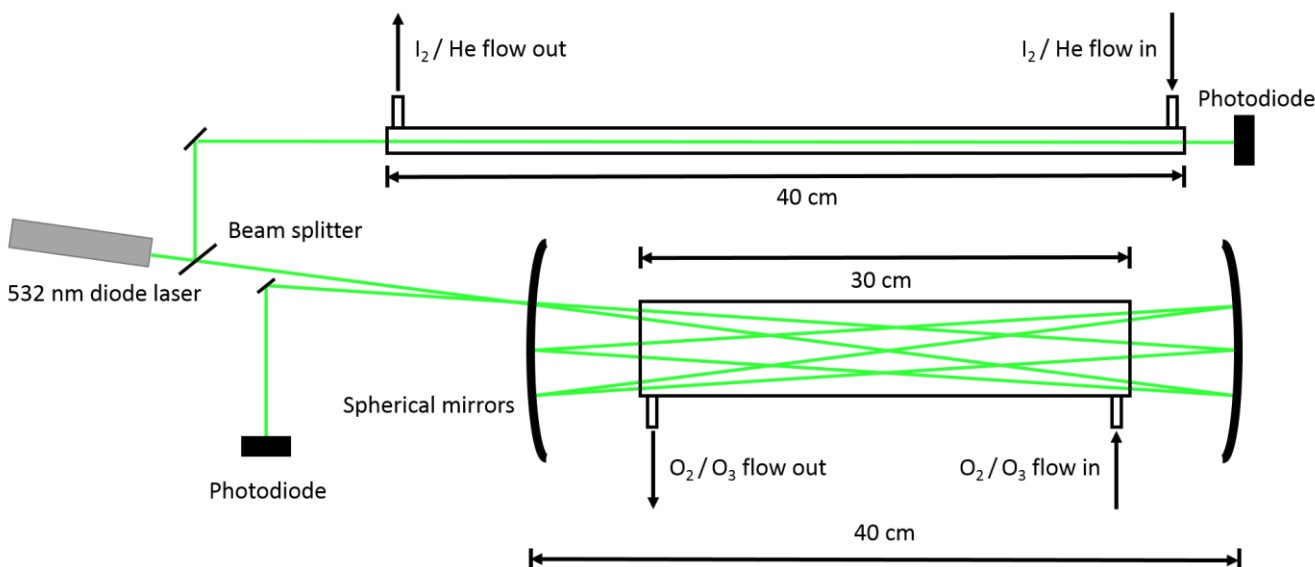

**Figure 2: A Schematic of the absorption setup used to measure both $I_2$ and $O_3$**

The concentrations $C$ of $O_3$ and $I_2$ were then determined using the Beer-Lambert law (equation 1), where $I$ and $I_0$ are the intensities of the 532 nm laser light recorded with and without the reagent present, respectively, $OD$ is the optical density, $\sigma$ the absorption cross section, and $l$ the optical path length:

$$OD = \ln\frac{I_0}{I} = \sigma \cdot C \cdot l \tag{1}$$

$O_3$ has an absorption cross-section at 532 nm of $2.34 \times 10^{-21}$ cm$^2$ molecule$^{-1}$ (Burrows et al., 1999). A typical concentration of $5 \times 10^{14}$ molecule cm$^{-3}$ in the reaction mixture (total flow ~1700 sccm) requires a concentration in the $O_3/O_2$ flow (100 sccm) of around $8.5 \times 10^{15}$ molecule cm$^{-3}$. An ozone concentration of $8.5 \times 10^{15}$ molecule cm$^{-3}$ results in $OD = 0.024$. $I_2$ has an absorption cross-section at 532 nm of $3.03 \times 10^{-18}$ cm$^2$ molecule$^{-1}$ (Saiz-Lopez et al., 2004). A typical $I_2$ concentration of $4 \times 10^{14}$ molecule cm$^{-3}$ in the reaction mixture (total flow ~1700 sccm) requires a concentration in the $I_2/He$ flow (100 sccm) of around $6.8 \times 10^{15}$ molecule cm$^{-3}$. An $I_2$ concentration of $6.8 \times 10^{15}$ molecule cm$^{-3}$ gives $OD = 0.82$. The ODs observed for both species are easily detectable by the instrument, which has a typical signal-to-noise ratio of ~ 400 , which corresponds to a minimum detectable OD of $2.5 \times 10^{-3}$. As there is no reference photodiode in this setup, drift in laser intensity must be accounted for. To negate the effect of laser drift as much as possible, the probe intensity is measured over a period of 15 seconds for both I and $I_0$, with ~10s between the measurements. To ensure that laser drift does not significantly affect the

measured concentrations, measurements are taken until 3 concordant results are obtained (typically the first three results).

In order to measure the photodepletion of the $I_xO_y$ molecules at 532 nm and 355 nm, a frequency-doubled or tripled Nd:YAG laser beam (Continuum Surelite 10-II, 1 cm$^{-1}$ linewidth, 20 ns pulse width, ~80 mJ pulse$^{-1}$ cm$^{-2}$ at 532 nm and ~50 mJ pulse$^{-1}$ cm$^{-2}$ at 355 nm) was passed across the flow tube, perpendicular to the main axis and near the sampling point. Experiments were carried out at 4-7 Torr. The pressure in the reactor was set by a throttle valve placed upstream of an Edwards 80 roots blower – oil rotary pump combination. The flows were set using MKS calibrated mass flow controllers and pressure was

measured using a set of 10 Torr and 1000 Torr MKS Baratron pressure transducers.

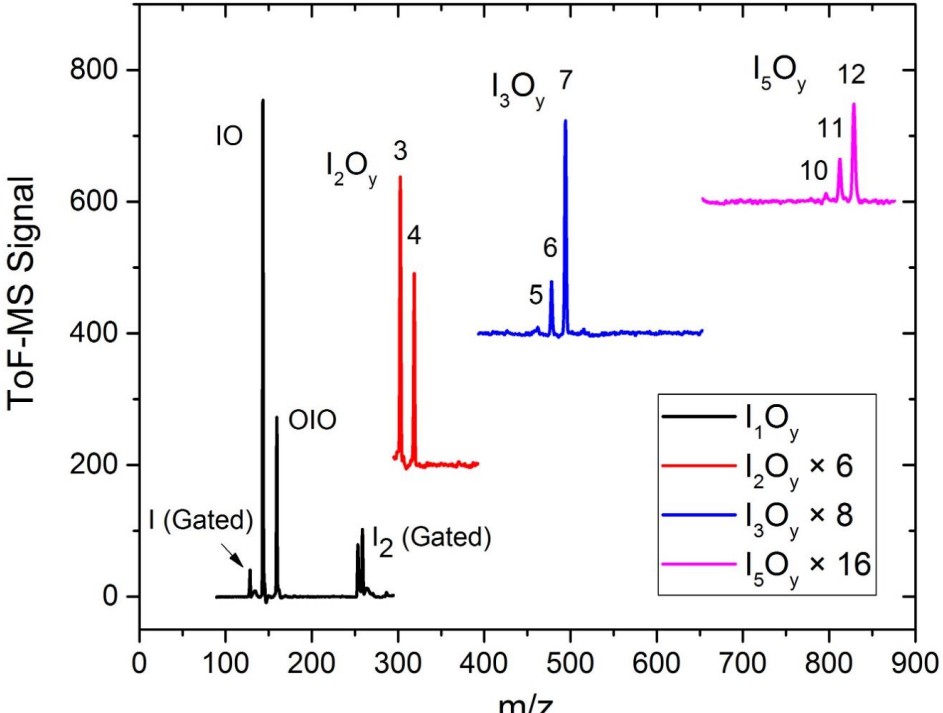

**Figure 3: Mass spectra in four different m/z ranges encompassing peaks of species with 1, 2, 3 and 5 iodine atoms. Some of the spectra are scaled and shifted for clarity. The mass-spectra are generated by averaging the mass spectra at each time point, over a 20 ms experiment comprising 20 individual time points.**

The iodine oxides generated were sampled in situ from the irradiated volume through a pinhole (200 μm diameter) situated on axis into a Kore Technology photo-ionization time-of-flight mass spectrometer (PI-ToF-MS) described in detail elsewhere (Gómez Martín et al., 2016). Successful detection of $I_xO_y$ by this method has been demonstrated elsewhere (Gómez Martín et al., 2013; Wei et al., 2017; Gómez Martín et al., 2020). The PI chamber of this instrument (~$10^{-4}$ Torr) is fitted with viewports,

allowing a pulsed laser beam to be directed through the high density region of the sampled gas jet. Iodine oxides have ionization potentials in the range of 9-11 eV (Gómez Martín et al., 2013; Wei et al., 2017), which requires generating a vacuum ultraviolet (VUV) laser ionization beam. This is achieved by tightly focusing the 355 nm output of a frequency-tripled Nd:YAG laser (Continuum Surelite 10-II pulse width 3-5 ns) in a cell filled with xenon, which produces VUV radiation by frequency tripling

(118 nm , or equivalently 10.5 eV) (Kung et al., 1973; Mahon et al., 1979). The resulting positive ions are accelerated towards

the ToF-MS by means of a continuous negative voltage. An electron photomultiplier detector coupled to a pre-amplifier outputs an analog signal, which is registered by the digital oscilloscope (Picoscope 6000). A limitation of this method is that large signals (e.g. $I_2^+$ and $I^+$) cause detector overload during a significant time span after the large peak has been registered. An analog gating box is used to lessen these effects by sending a (~ 400 ns wide) gating pulse to the ion extraction optics. Gating in this way removed >90% of the overloaded signal from $I_2^+$ and $I^+$, significantly improving the signal to overload ratio.

Synchronization between the chemistry-initiating excimer laser pulse, the photolysis laser, the probing PI laser pulse and the detection devices is provided by a computer-controlled delay generator (Quantum Composers, 9518). In this manner, the delay between PLP and PI can be varied in order to observe the kinetics of reactants and products, and the photodepletion of the species of interest. The experiment repetition rate was set at 10 Hz, the optimal repetition rate for operation of both the VUV generating PI YAG laser and the YAG laser used for photolysis of the iodine oxides. A LabView program built in-house

provides a sequence of delays between the PLP and PI lasers which can be modified by the user via a graphical interface. Desired experimental parameters are input into the LabView program, including the number of pre-photolysis points, the number of experimental data, the duration of the experiment and the number of repetitions. Each experiment results in a 3-dimensional dataset of signal intensity (proportional to concentration) vs. 248 nm photolysis - VUV photoionization delay time (kinetic profile or time trace) and time-of-flight (mass spectrum). Figure 3 shows mass spectra with the most prominent

peaks obtained at different delay times. Mass-to-charge ($m/z$) calibration of time-of-flight was performed by selecting a number of well-known prominent mass peaks (e.g. IO at $m/z$ =143, OIO at $m/z$ =159, $I_2O_3$ at $m/z$ =302, $I_2O_4$ at $m/z$ =318 and $I_3O_7$ at $m/z$ =493 (Gómez Martín et al., 2013)).

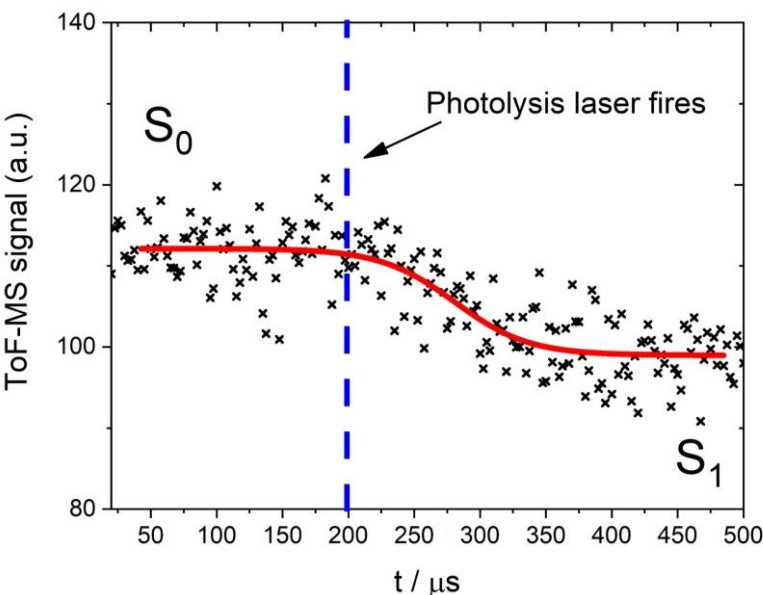

**Figure 4: Schematic of a photodepletion measurement of NO₂. The red curve is an empirical fit through the data performed to obtain the values of S₀ and S₁.**

To investigate the photodepletion of $I_xO_y$, it is desirable to keep the signal of the species of interest relatively constant over the delay window used to probe the photolysis. The timing of the Nd:YAG photolysis laser pulse was programmed to a fixed delay with respect to the 248 nm excimer pulse. To find the optimal window in which to conduct photolysis experiments, the reaction kinetics were first explored between 0 and ~10 ms for different sets of conditions. It was important to generate a sufficiently high concentration of $I_xO_y$ to obtain a good signal-to-noise ratio, but not so high that the growth/removal timescale was comparable to the observed photodepletion (~400 µs).

The trace shown in Figure 4, which is similar to all the traces obtained in this experiment, exhibits a delay between the pre- and post-photodepleted signal. This delay corresponds to an instrumental sampling time depending on the alignment of the lasers and the molecular masses of the bath gas and the sample species (Baeza-Romero et al., 2012). In the present experiments, it was necessary to leave a small gap (~2 mm) between the photolysis volume and the sampling pinhole, so as to avoid hitting the skimmer cone with the laser. The diffusional exchange of molecules between the photodepleted volume, and the un-photodepleted volume immediately before the pinhole blurs the onset of the photodepletion as it is measured by the TOF-MS. The values of $S_0$ and $S_1$ were obtaining by empirically fitting the photodepletion trace to a sigmoidal function:

$$y = S_0 + \frac{S_1 - S_0}{1 + 10^{(\log x_0 - x)p}} \tag{2}$$

Fitting to this function ensures that the flat sections corresponding to the pre- and post-photodepletion concentrations ($S_0$ and $S_1$ respectively) are characterised in the precise regions outside of the aforementioned "blurred" zone. The parameters $x$ and $p$ are not of scientific importance for this study and are simply instrumental factors.

The photolysis laser depletes any analyte within the photolysis volume which exhibits a bound-unbound transition at the energy of the photolysis wavelength used for each experiment. The signals for a certain $m/z$ ratio before and after the photolysis laser is fired are respectively noted as $S_0$ and $S_1$ (Figure 4). Absorption cross sections were then calculated by comparing the relative signal change ($\zeta_{IxOy}$) for the analyte of interest to the relative signal change ($\zeta_{REF}$) for a molecule having a well-studied spectrum in an analogous actinometric experiment using the same configuration and laser fluence. In this way, systematic errors arising from the beam energy and area are eliminated in the determination of the actinic flux $F$:

$$\frac{S_{REF\,0} - S_{REF}}{S_{REF\,0}} = \frac{[X_{REF}]_0 - [X_{REF}]}{[X_{REF}]_0} \equiv \zeta_{REF} = 1 - e^{-F \cdot \sigma_{X_{REF}} \cdot \phi_{X_{REF}}} \tag{3}$$

$$F = -\frac{1}{\sigma_{X_{REF}} \cdot \phi_{X_{REF}}} \ln(1 - \zeta_{REF}) \tag{4}$$

Similarly, for the target $I_xO_y$ molecule we have:

$$\frac{S_{IxOy\,0} - S_{IxOy}}{S_{IxOy\,0}} = \frac{[I_xO_y]_0 - [I_xO_y]}{[I_xO_y]_0} \equiv \zeta_{I_xO_y} = 1 - e^{-F \cdot \sigma_{I_xO_y} \cdot \phi_{I_xO_y}} \tag{5}$$

$$\sigma_{I_xO_y} \cdot \phi_{I_xO_y} = -\frac{1}{F} \ln(1 - \zeta_{I_xO_y}) \tag{6}$$

Inserting equation 4 in equation 6, we have:

$$\sigma_{I_xO_y} \cdot \phi_{I_xO_y} = \sigma_{X_{REF}} \phi_{X_{REF}} \frac{\ln(1-\zeta_{I_xO_y})}{\ln(1-\zeta_{REF})} \tag{7}$$

$NO_2$ (99.5%, BDH, 10% in He) was the best reference molecule to perform actinometry at 355 nm. In this case, $\sigma_{X_{REF}} = \sigma_{NO_2(355\,nm)} = 5.3 \times 10^{-19}$ cm$^2$ molecule$^{-1}$ (Vandaele et al., 1998) and $\phi_{X_{REF}} = \phi_{NO_2(355\,nm)} = 1$ (Burkholder et al., 2015).

Typically, $NO_2$ depletion was around 8% in the actinometric experiments. For the 532 nm photolysis experiments, OIO was chosen as reference, owing to the low sensitivity of its broad absorption bands to photolysis laser resolution, as opposed, for example, to $I_2$. The use of $I_2$ as actinometer is also precluded by the overload issue mentioned above. The OIO absorption cross section at 532 nm is relatively well known, within 25% of the average value of the four independent determinations reported in the literature (Bloss et al., 2001; Joseph et al., 2005; Spietz et al., 2005; Tucceri et al., 2006). On the other hand,

conflicting results have been reported for the OIO photolysis quantum yield (Tucceri et al., 2006; Gómez Martín et al., 2009). Here we use the unit quantum yield reported by Gomez Martin et al. (2009), which was determined in a system free of interferences from ozone where a long-lived I atom photofragment and no reformation of OIO was observed over a time scale of several milliseconds. This result is also supported by the short lifetime (200 fs) of the excited state responsible for the observed absorption bands (Ashworth et al., 2002) and the existence of a feasible photolysis path revealed by high level ab

initio calculations (Peterson, 2010). Thus $\sigma_{X_{REF}} = \sigma_{OIO(532\,nm)} = 9.5 \times 10^{-18}$ cm$^2$ molecule$^{-1}$ (Bloss et al., 2001; Spietz et al., 2005) and $\phi_{X_{REF}} = \phi_{OIO(532\,nm)} = 1$ (Gómez Martín et al., 2009). In principle, equation 7 yields the photolysis cross section, i.e. the product $\sigma_{I_xO_y} \cdot \phi_{I_xO_y}$, which is equal to the absorption cross section if $\phi_{I_xO_y} = 1$. This is a reasonable assumption for broad band absorption spectra indicating excitation to an unbound upper state. The photolysis processes in this study therefore do not depend on the nature and pressure of the carrier gas matrix, and the results can be applied directly in atmospheric

models.

### 2.2. Ab initio calculations

The geometry of the ground electronic state of $IO_3$, $I_2O_3$, $I_2O_4$, $I_3O_6$ and $I_3O_7$ was optimised by using the B3LYP functional combined with the standard 6-311+G(2d,p) triple-$\zeta$ basis set for O and H, and an all-electron (AE) basis set for I (Glukhovtsev et al., 1995). The AE basis set is a contracted (15s12p6d)/[10s9p4d] 6-311G basis, the [5211111111,411111111,3111]

contraction scheme supplemented by diffuse s and p functions, together with d and f polarization functions. See Glukhovtsev et al. (1995) for further details. Figure A1 illustrates the geometries of the various iodine oxides and hydroxy- species that are discussed in Section 4. The corresponding molecular parameters (Cartesian coordinates, rotational constants and vibrational frequencies) are listed in Table A1. Note that the ground states some of these oxides have been studied at a higher level of theory elsewhere (Kaltsoyannis and Plane, 2008; Galvez et al., 2013).

The absorption cross-sections of the aforementioned molecules were calculated using the time-dependent density functional (TD-DFT) excited states method (Stratmann et al., 1998) within the Gaussian 16 suite of programs (Frisch et al., 2016). The vertical excitation frequencies ($\tilde{\nu}_i$) and oscillator strengths ($f_i$) were obtained for the first 30 excited states (Table S2). The

cross section $\sigma(\tilde{v})$ at frequency $\tilde{v}$ was then computed using the GaussView program (Dennington et al., 2016):

$$\sigma(\tilde{v}) = 2.17 \times 10^{-16} \quad \sum_{i=1}^{n} \left( \frac{f_i}{\Delta} \exp\left[ -\left( \frac{\tilde{v} - \tilde{v}_i}{\Delta} \right)^2 \right] \right) \tag{8}$$

where the summation is over the 30 electronic excited states and $\sigma(\tilde{v})$ is in units of $cm^2$ molecule$^{-1}$. Each peak is assumed to have a Gaussian band shape with a width $\Delta$, set here to the default value (Dennington et al., 2016) of 0.4 eV. Although there are more advanced methods for the calculation of electronic spectra, TD-DFT offers a reasonable compromise between low computational cost and accuracy of the predicted transitions. Figure A2 shows a comparison between the experimental and TD-DFT absorption spectra of IO, OIO and HOI. Note that although the TD-DFT method is not designed to predict ro-

vibrational structure, the spectral positions of the electronic bands and the average absorption cross sections are in reasonable agreement with the experiment, even in the case of open shell species like IO and OIO. Higher iodine oxides are closed shell molecules and the accuracy of the transitions is expected to be similar to the result for HOI.

## 2.3. Photolysis rate calculations

In this study we employ the global 3D model CAM-Chem (Community Atmospheric Model with chemistry, version 4.0),

included in the CESM framework (Community Earth System Model) (Lamarque et al., 2012), to estimate the photolysis rate ($J$) of the different $I_xO_y$ species according to their computed absorption cross-section, constrained by the experimental data at 532 and 355 nm. CAM-Chem has been configured with a 2.5º longitude by 1.9º latitude spatial resolution and 26 vertical level (from the surface to up to 40 km). The model was run in the specified dynamics mode (Lamarque et al., 2012), using offline meteorological fields from a previous free-running climatic simulation (Fernandez et al., 2014; Saiz-Lopez et al., 2015).

CAM-Chem  implements a state-of-the-art halogen chemistry scheme (Fernandez et al., 2014; Saiz-Lopez et al., 2014; Saiz-Lopez et al., 2015). This chemical scheme is fed by both organic and inorganic halogen emission sources (Ordóñez et al., 2012; MacDonald et al., 2014). Saiz-Lopez and co-workers ((Saiz-Lopez et al., 2014), hereafter SL2014) conducted two sets of simulations without and with the photolysis of $I_xO_y$ oxides in CAM-Chem to evaluate the range of inorganic iodine loading, partitioning and impact in the troposphere. The photolysis of $I_xO_y$ was based on the best available knowledge at that time on

the major $I_xO_y$ species ($I_2O_2$, $I_2O_3$ and $I_2O_4$) and their absorption cross sections (Gómez Martín et al., 2005; Gómez Martín et al., 2013). In this study we run photolysis simulations and compare the $J$ values with the new cross sections to those in SL2014.

## 3. Results

### 3.1. Photolysis at 355 nm

As described in the experimental section, kinetic profiles of the growth and removal of the target iodine oxide species shown

in Figure 3 were carried out in order to define the time periods with the most suitable kinetic profiles for photolysis

measurements (Figure 5). Fragmentation of iodine oxides was a significant problem in these experiments, as predicted from ionization energy calculations of larger iodine oxides to possible photofragments at 10.5 eV(Gómez Martín et al., 2020). High amounts of active iodine ($IO_x$ = I, IO) released from reaction R12 lead to fast formation of $I_xO_y$ and particles. Under these conditions, at long times after the peak IO and OIO (~3-5 ms), the observed signal of IO, OIO and $I_xO_y$ is contaminated by

photofragmentation of higher order iodine oxides. For this reason, great care needed to be taken to establish a time window for each species wherein higher oxides are not present, to ensure that any depletion in the mass spectrometric signal for each species is exclusively due to the removal of the species via photolysis (Figure B1). Evidence of fragmentation comes in the form of a secondary growth in the signal seen for IO and OIO (Figure 5). The delay between the excimer and the Nd:YAG photolysis laser was therefore carefully selected to coincide with a period of relatively constant signal of the desired analyte,

typically a maximum for short lived species, or a slow rise for larger reaction products of interest.

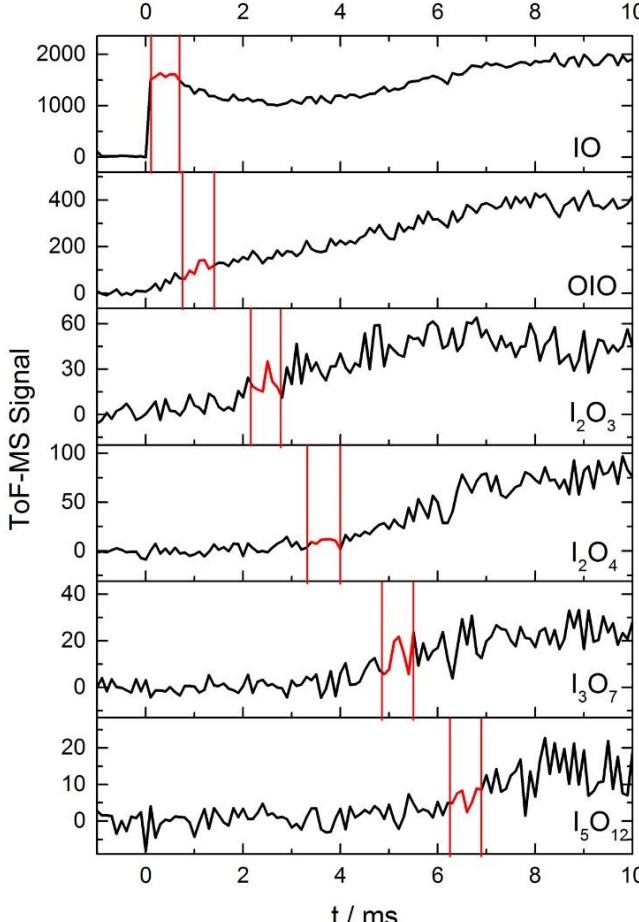

**Figure 5. Time traces of IO, OIO, $I_2O_3$, $I_2O_4$, $I_3O_7$ and $I_5O_{12}$ from -1-10 ms, at 1 ms intervals for a mixture of He (10 torr), $O_3$ and $I_2$ ([$O_3$] = 4 × 10$^{14}$ molecule cm$^{-2}$ [$I_2$] = 2.8 × 10$^{14}$ molecule cm$^{-2}$) flash photolyzed by an excimer pulse at t = 0 (130 mJ pulse$^{-1}$). The red sections highlighted for each species correspond to the optimal delay windows for photolysis of the corresponding species for this set**

**of conditions.**

In the case of IO and OIO, the photolysis delay was selected based on the location of the maxima of their kinetic profiles. Typically, this was around 1 and 2 ms for IO and OIO respectively. IO showed a clear depletion of ~20% at 355 nm (Figure 6a) corresponding to a cross section of $\sigma_{355\,nm,\,IO} = (1.2 \pm 0.1) \times 10^{-18}$ cm$^2$ molecule$^{-1}$ (based on an average of 3 measurements) in excellent agreement with the most recent determinations based on absorption spectroscopy (Bloss et al., 2001; Spietz et al.,

2005), where the presence of an underlying absorption corresponding to a product of reactions R2, R4, R5 or R6 was accounted for (Atkinson et al., 2007) (Figure 6b). We consider the mutual consistency between our measurement and the most reliable determinations of the IO cross section at 355 nm as a validation of our method for determining the absorption cross sections of $I_xO_y$ species.

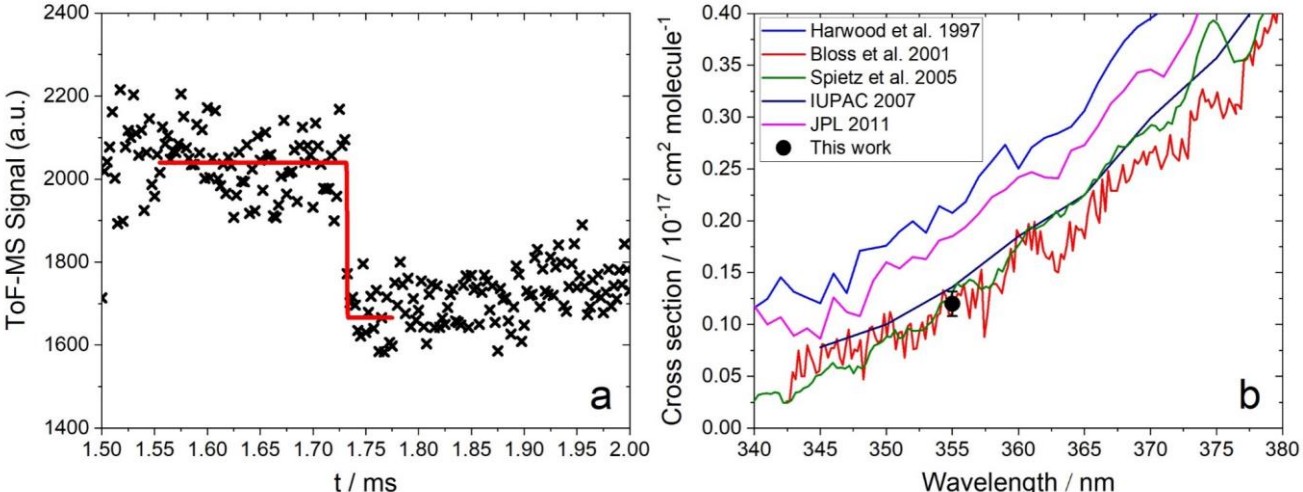

**Figure 6: Panel a: measured photodepletion of IO (m/z = 143). Panel b: comparison of published absorption cross section with the absorption cross section at 355 nm determined in this work**

As expected, no significant depletion was observed for OIO, with an upper limit of $\sigma_{355\,nm,\,OIO} < 10^{-19}$ cm$^2$ molecule$^{-1}$ based on

the noise of the mass spectrometric signal at $m/z$ 159. No significant depletion was observed for $I_2O_3$ ($m/z = 302$) at 355 nm (Figure 7, top left), and an upper limit for the absorption cross section of $\sigma_{355\,nm,\,I2O3} < 5 \times 10^{-20}$ cm$^2$ molecule$^{-1}$ was determined based on the average noise of the $I_2O_3$ signal. By contrast, $I_2O_4$ was depleted by 355 nm photolysis by ~40% (Figure 7, top right), corresponding to a photolysis cross-section of $\sigma_{355\,nm,\,I2O4} = (2.7 \pm 0.3) \times 10^{-18}$ cm$^2$ molecule$^{-1}$ (based on an average of 2 measurements).

The molecular clusters $I_3O_7$ ($m/z = 493$) and $I_5O_{12}$ (827) are two of the major peaks of successive mass peak progressions observed in the mass spectra in Figure 3. Experimental runs with wider spectral windows show peaks that correspond to even larger molecular clusters: $I_7O_{17}$ (1161), $I_9O_{22}$ (1495), $I_{11}O_{27}$ (1829) and $I_{13}O_{32}$ (2163). Notably, no mass peaks with even number of iodine atoms appear, even for higher pressures and using $N_2$ as carrier gas (Gómez Martín et al., 2020). It is also interesting that these peaks are separated by 334 $m/z$ units corresponding to $I_2O_5$. Although $I_2O_5$ is not observed in the gas phase, it is

noted that as the clusters grow larger, the ratio O:I ratio tends to 2.5, reaching 2.46 for $I_{13}O_{32}$, i.e. within the range of variability of the O:I ratio of iodine oxide particles (IOPs) determined by EDX (O/I = 2.42 ± 0.05) (Saunders et al., 2010). The prominent

peak at $m/z = 493$ is the largest mass observed in our previous work on $I_xO_y$ (Gómez Martín et al., 2013) and appears in all the spectra acquired in the course of the present work. From the observed photodepletion of $I_3O_7$ at 335 nm we have determined an absorption cross section of $\sigma_{355\,nm,\,I3O7} = (5.3 \pm 1.4) \times 10^{-18}$ cm$^2$ molecule$^{-1}$ (based on an average of 3 measurements) (Figure 7). The measured cross section for the $I_5O_{12}$ cluster is $\sigma_{355\,nm,\,I5O12} = (9.8 \pm 1) \times 10^{-18}$ cm$^2$ molecule$^{-1}$ (based on an average of 2 measurements). The determined cross sections and upper limits at 355 nm are listed in Table 1.

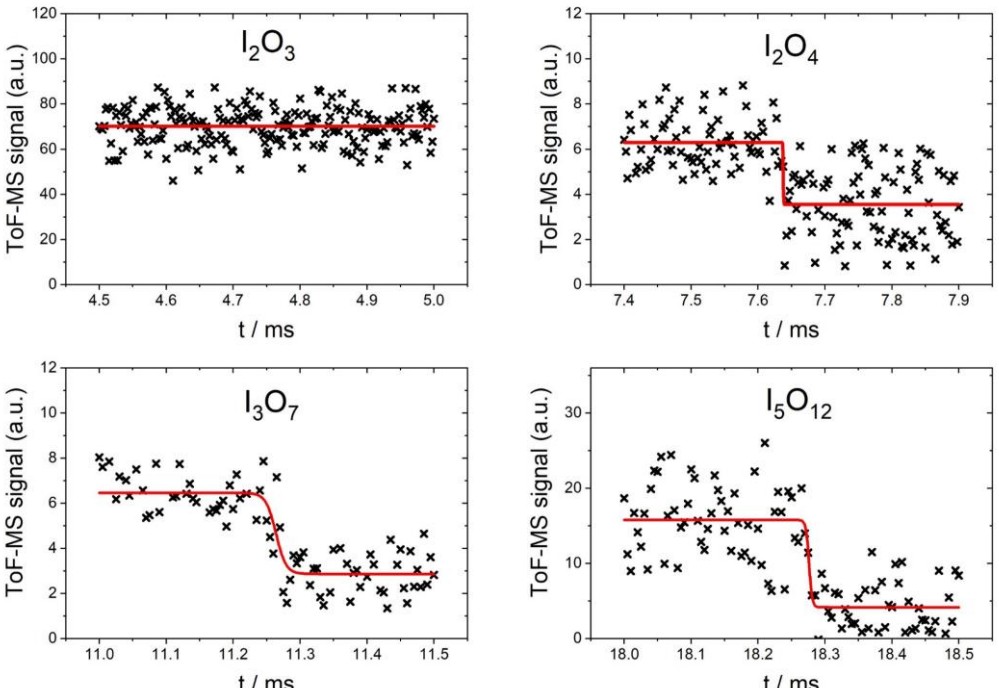

**Figure 7:** **Measured photodepletion at 355 nm of $I_2O_3$, $I_2O_4$, $I_3O_7$ and $I_5O_{12}$. The laser was triggered at 4.75 ms in the $I_2O_3$ experiment. The red line indicates the fit to equation 2.**

Fragmentation precludes determination of photolysis products at 355 nm (channels of photochemical reactions R9-R11) due to the fact that the signal arising from precursor ions (which may be 355 nm photolysis products), is indistinguishable from product ions resulting from fragmentation of higher oxides due to the 118 nm photoionization. For any given species, possible increases in signals corresponding to products of photolysis at 355 nm are offset by the decrease in the precursor ion concentration and therefore the product ion signal. As an example, consider $I_3O_7$ photolysis at 355 nm. The peaks of the possible photolysis products $I_2O_3$, $I_2O_4$, OIO and IO (referred to here as photofragments), prior to photolysis will have contributions from the precursor ion signals of these species, as well as product ion signals of $I_3O_y$. Upon photolysis of $I_3O_7$, the concentration of $I_3O_7$ will decrease, causing a decrease in the precursor ion signal, and therefore the product ion signals, resulting in turn in a net negative impact on the amplitude of all product ion signals. In addition to the decrease in product ion signals resulting from possible photofragment species, the precursor ion signal for the photolabile species (IO, $I_2O_4$ in this example) will decrease due to direct photolysis of these species. Concurrently with the decrease in the $I_3O_7$ precursor ion

signal, there will be a positive contribution to some or all of the concentrations of possible photofragment precursor ion signals. It is impossible to separate these contributions to the net change in signal magnitude, and therefore the products of these photolysis processes cannot be determined.

As indicated by reactions R10b and R11b, $IO_3$ ($m/z$ = 175) is a potential product of $I_2O_3$ and $I_2O_4$ photolysis. However, this species has never been observed in the gas phase. According to our ab initio calculations, the ionisation potential of $IO_3$ is 12.1 eV, which is higher than the photoionization energy employed in the present work (10.5 eV).

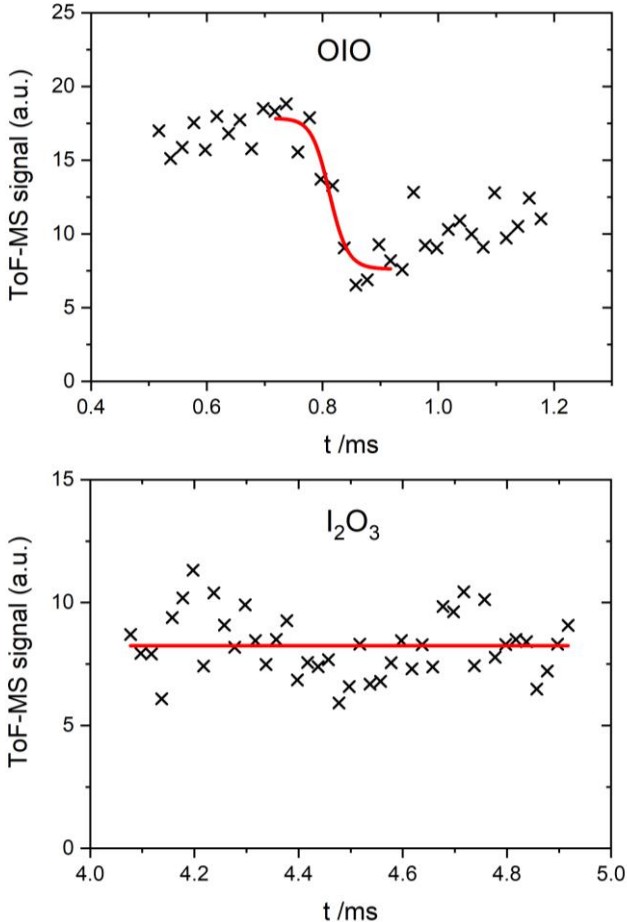

**Figure 8. Measured photodepletion at 532 nm of OIO and $I_2O_3$. The laser was triggered at 4.5 ms in the $I_2O_3$ experiment.**

### 3.2. Photolysis at 532 nm

As expected (Gómez Martín et al., 2009), OIO shows strong depletion in the 532 nm photolysis experiments (Figure 8), and it is therefore used as the actinometer at this wavelength. None of the higher order iodine oxides showed measurable depletion above the noise (as an example the $I_2O_3$ trace is shown in Figure 8). Table 2 lists the upper limits to the absorption cross-sections of each species at 532 nm, determined from the signal-to-noise ratio of the recorded traces.

**Table 1: Absorption cross sections of iodine oxides at 532 nm and 355 nm**

| Species | $\sigma_{355\,nm}$ /cm$^2$ molecule$^{-1}$ | $\sigma_{532\,nm}$/cm$^2$ molecule$^{-1}$ |
|---|---|---|
| IO | $(1.2 \pm 0.1) \times 10^{-18}$ | [a] |
| OIO | $< 10^{-19}$ | [b] |
| $I_2O_3$ | $< 5 \times 10^{-19}$ | $<5.0 \times 10^{-19}$ |
| $I_2O_4$ | $(3.9 \pm 1.2) \times 10^{-18}$ | $<1.3 \times 10^{-18}$ |
| $I_3O_6$ | $(6.1 \pm 1.6) \times 10^{-18}$ | $<1.5 \times 10^{-18}$ |
| $I_3O_7$ | $(5.3 \pm 1.4) \times 10^{-18}$ | $<1.4 \times 10^{-18}$ |
| $I_5O_{12}$ | $(9.8 \pm 1.0) \times 10^{-18}$ | $<1.5 \times 10^{-18}$ |

305    [a] The 0←0 band of IO is placed at 465.5 nm (Spietz et al., 2005), and therefore, there cannot be any band of ground state IO ($v'' = 0$) at a longer wavelength. There is a hot band at ~530 nm (0←4), but the $v'' = 4$ state is not populated at room temperature in equilibrium.
[b] Actinometer

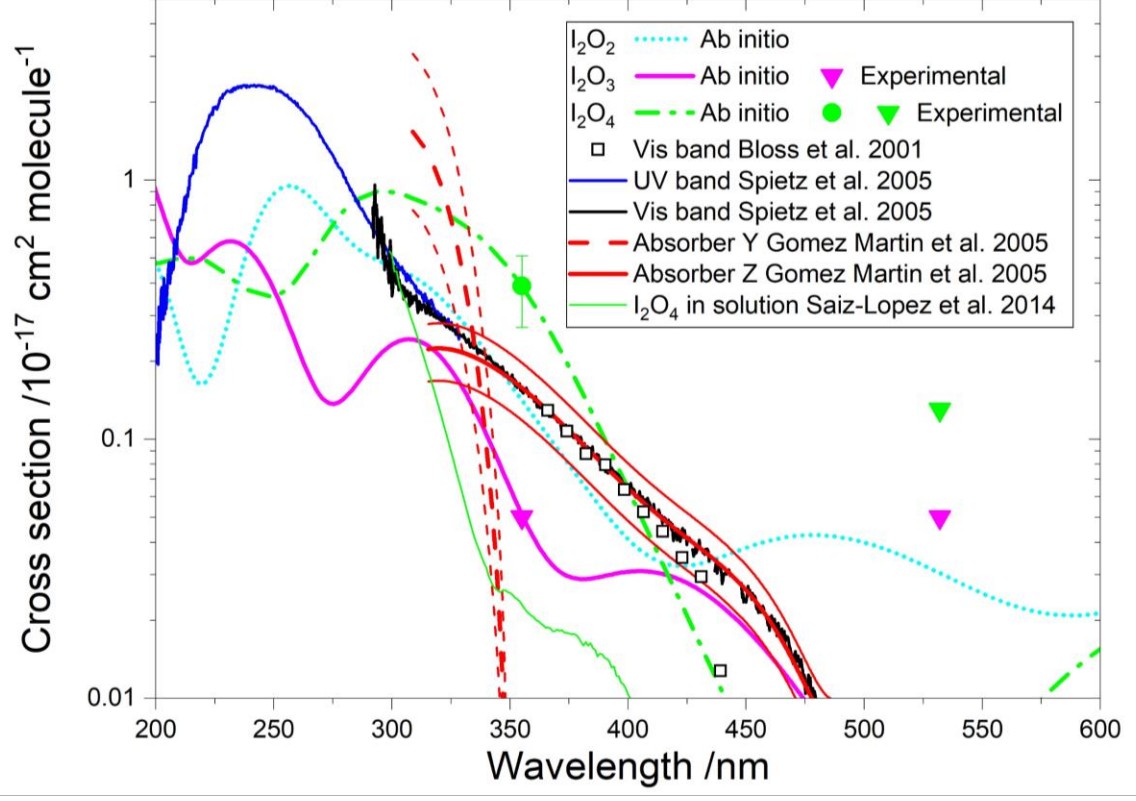

**Figure 9: Absorption cross sections of I₂O₂, I₂O₃ and I₂O₄. The cyan, pink and green lines correspond to the respective ab initio spectra. The pink and green symbols indicate respectively the measured absorption cross sections of I₂O₃ and I₂O₄ (triangles indicate upper limits). The empty squares correspond to an absorption band reported by Bloss et al. (2001) in 8 nm interval averages for λ > 362 nm. The blue and black lines correspond to absorption bands reported by Spietz et al. (2005). The latter was found to be a superposition of two absorptions with different kinetic behaviour, denoted Y (dashed red line) and Z (solid red line) (Gómez Martín et al., 2005). The thin red lines represent the uncertainties of these spectra, which mainly originate from the absolute absorption cross sections of Y and Z at 322 and 356 nm respectively. A spectrum of I₂O₄ in solution is indicated in green (Saiz-Lopez et al., 2014).**

## 3.3. Ab initio spectra

The calculated spectra are displayed in Figure 9 ($I_2O_2$, $I_2O_3$ and $I_2O_4$), Figure 10 ($I_3O_6$ and $I_3O_7$), and Figure 11 ($IO_3$). Oscillator strengths of the electronic transitions that are responsible for the visible and UV absorptions are provided in Appendix A. The TD-DFT spectra were wavelength-shifted by applying a constant energy shift to get agreement with the experiment at 355 nm. The shifts are quite modest, within the expected error at this level of theory (Foresman and Frisch, 2015): $I_2O_3$ (30 kJ mol$^{-1}$), $I_2O_4$ (-12 kJ mol$^{-1}$), $I_3O_6$ (9.2 kJ mol$^{-1}$), $I_3O_7$ (-21 kJ mol$^{-1}$). Applying a constant energy shift means assuming that all the excited state energies are offset by a constant amount with respect to the ground state.

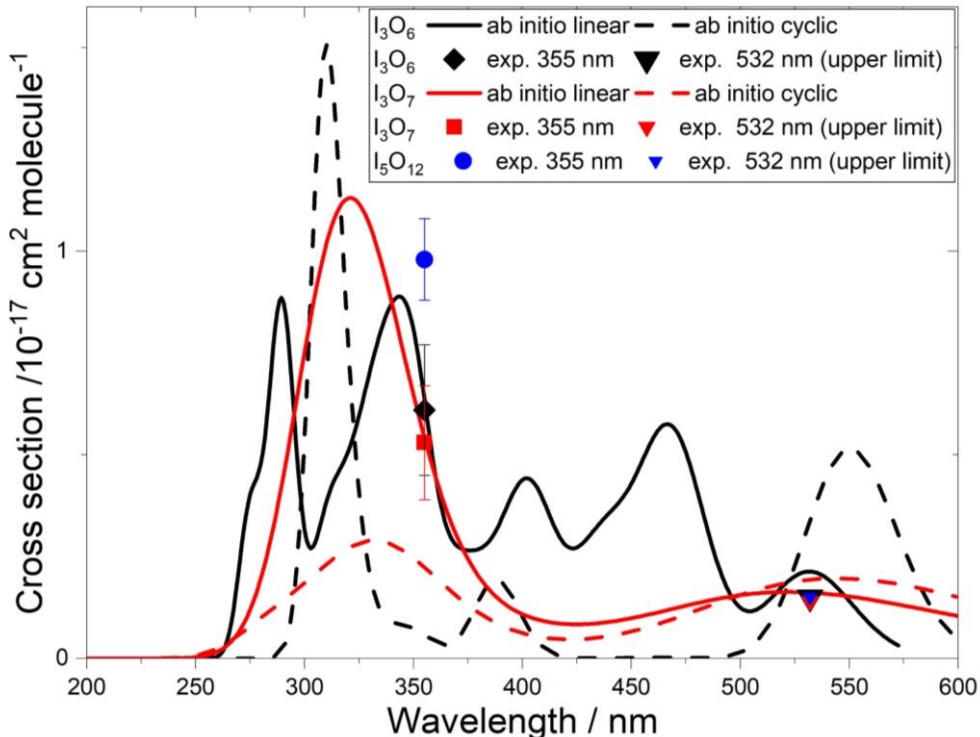

**Figure 10: Absorption cross section spectra of $I_3O_6$ (black lines and symbols), $I_3O_7$ (red lines and symbols) and $I_5O_{12}$ (blue symbols). Ab initio spectra of linear and cyclic isomers are shown by solid and dashed lines, respectively.**

## 4. Discussion

### 4.1. Absorption spectra

A time-dependent broad band absorption was observed by Bloss et al. (2001) concurrently with the IO self-reaction decay, which was assigned to $I_2O_2$. The shape of the band was observed to be different for $\lambda < 360$ nm depending on the chemical

scheme employed to generate iodine oxides. Later, a broad band absorption with a similar spectral slope was reported (Figure

9) together with targeted UV measurements showing that the band peaks around 250 nm (Gómez Martín et al., 2005; Spietz et al., 2005). The kinetics of the absorption spectrum for $\lambda > 310$ nm were found to result from the overlap of at least two different species, one dominating the absorption below 340 nm (labelled as Y) and another one dominating from 340 nm towards the visible (labelled as Z). It was found that the kinetics of species Y was compatible with a product of the IO self-reaction, i.e. $I_2O_2$. By contrast Z, which would be similar to the spectrum measured by Bloss et al. for $\lambda > 360$ nm, was not compatible with a product of the IO self-reaction, and it was tentatively assigned to $I_2O_3$.

Gomez Martin et al. (2005) derived the cross section of the species Y and Z at 322 nm and 356 nm respectively, using an iodine mass conservation approach where the depletion of the $I_2$ precursor was balanced by the sum of atomic iodine contained in the oxides formed after the photolysis pulse. The cross sections were determined in a per iodine atom basis. Assuming absorbers with two iodine atoms, the spectra of Y and Z were scaled as shown in Figure 9 (Saiz-Lopez et al., 2014). Since the spectra are featureless, they can be assumed to result from bound to free transitions, justifying a unit photolysis quantum yield of the molecules generating them.

The absorption cross sections determined in this work from photodepletion of $I_2O_3$ and $I_2O_4$ at 355 nm do not confirm the previous tentative assignment of absorber Z to $I_2O_3$. The cross section of $I_2O_4$ obtained from our photodepletion experiments is in better agreement with the absorber Z. Also, it appears that the spectrum of $I_2O_4$ in solution largely underestimates the gas-phase spectrum. The absorption of Z at 355 nm in Figure 9 lies slightly below the $I_2O_4$ cross section determined in the present work, which is consistent with a contribution to Z of other $I_xO_y$ species with $x > 2$ (i.e. the scaling of the spectrum should be effectively higher than the factor of 2 used by Gomez Martin et al. (2005)). In fact, the theoretical spectra of $I_2O_4$, $I_3O_6$ and $I_3O_7$ (Figure 9 and Figure 10) show similar shapes between 300 and 500 nm. It is very likely that the absorption band extending from 200 nm to nearly 500 nm (blue and black lines in Figure 9) results from the overlap of absorptions corresponding to several $I_xO_y$ species. Therefore, these experimental absorption spectra are not useful for photolysis rates calculations unless further deconvolution enables separation of the different contributions.

The calculated spectra shown in Figure 9 and Figure 10 agree generally well with the experimentally determined values at 355 nm and 532 nm after small wavelength shifts indicated above. As shown in Figure 10, the upper limits from experiment are within 20% of the theoretical calculations for linear $I_3O_6$ and $I_3O_7$. Note that theoretical absorption cross section spectra for both the linear and cyclic forms (Figure 5) of $I_3O_6$ and $I_3O_7$ are shown in Figure 10. The linear forms of both species fit significantly better to the experimental values, suggesting the linear form is the one which is formed preferentially in these reactions.

### 4.2. Photolysis products

The likely photodissociation pathways of $I_2O_3$ and $I_2O_4$ can be elucidated by seeing how the geometry of each excited state relaxes after vertical excitation from the ground-state geometry using the TD-DFT method (see Section 2.2). In the case of

I$_2$O$_3$, excitation in both the bands at 334 and 453 nm (Figure 9) produces a pronounced extension of the I–IO$_3$ bond, indicating photolysis leading to I + IO$_3$ (R10b). For I$_2$O$_4$, excitation at 667 nm and in its strong band at 324 nm produces IO + IO$_3$, whereas absorption in the weak band at 366 nm produces OIO + OIO.

The theoretical absorption spectrum of IO$_3$ is shown in Figure 11, calculated at the same level of theory as described in Section 2.2. IO$_3$ absorbs in the tropospheric solar actinic range in two near-IR bands centered at 710 and 950 nm. There are two possible photolysis pathways:

$$IO_3 + hv \rightarrow OIO + O \tag{R15}$$

$$\rightarrow IO + O_2 \tag{R16}$$

The first channel requires 180 kJ mol$^{-1}$ to break the O$_2$I–O bond (not including spin-orbit coupling), corresponding to a photolytic threshold of 665 nm.  R16 is closely analogous to the photolysis of OIO in its band between 500 and 600 nm, where the I atom is squeezed out from between the two O atoms in a process that overall is actually exothermic because of the strength of the newly-formed O$_2$ bond (Gómez Martín et al., 2009). In the same way, R16 is exothermic by 78 kJ mol$^{-1}$. However, there is a barrier of 145 kJ mol$^{-1}$ involved in squeezing the IO out from between the 2 oxygen atoms. This corresponds to a photolytic threshold of 827 nm. Both of these thresholds are indicated on Figure 11. This shows that absorption across the entire 710 nm band is sufficient for photolysis via R16 to occur. If that were the case, then $J(IO_3 \rightarrow IO + O_2)$ could be as large as 0.15 s$^{-1}$.

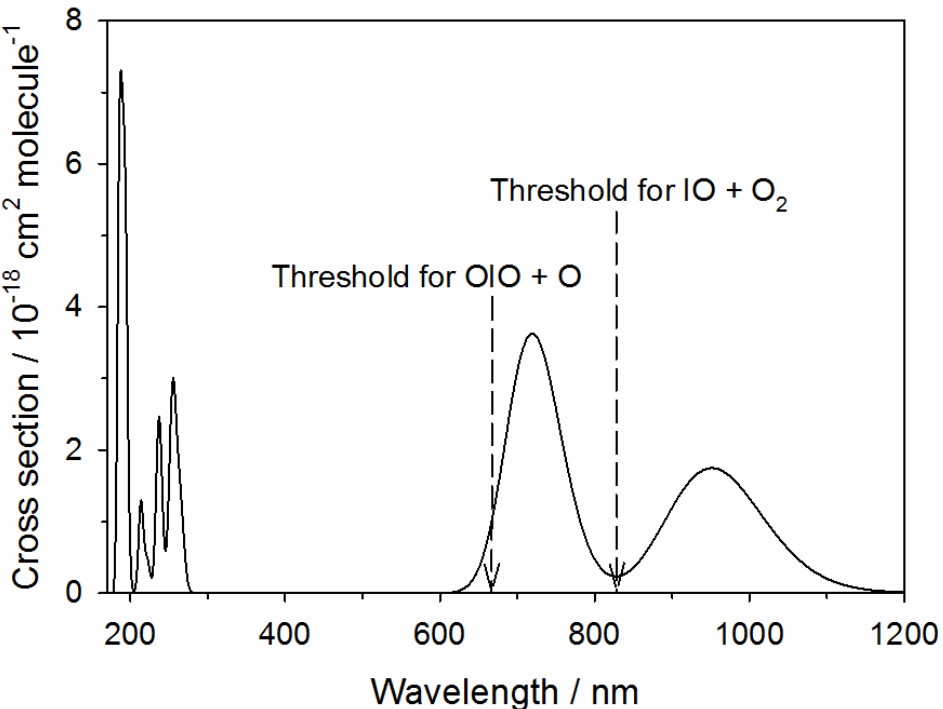

**Figure 11: Absorption spectrum of IO$_3$ calculated at the TD/B3LYP/gen level of theory (see text for further details). The arrows indicate the energy thresholds for the two photodissociation pathways.**

Photolysis via R15 is just possible in the short wavelength tail of the 710 nm band, so $J(IO_3 \rightarrow OIO + O)$ could be as large as 0.007 s$^{-1}$. It therefore seems likely that photolysis of IO$_3$ is fast and mainly produces IO.

If, however, photolysis of IO$_3$ is significantly slower than this upper limit, the molecule could persist in the atmosphere long enough to undergo reaction with other molecules. An intriguing possibility is that IO$_3$ may be a source of HIO$_3$. The reaction

370     IO$_3$ + H$_2$O → HIO$_3$ + OH                                                 (R16)

is endothermic by 25 kJ mol$^{-1}$ (14 kJ mol$^{-1}$ according to previous work at higher level of theory (Khanniche et al., 2017b)), and so most likely does not occur. However, the reaction

IO$_3$ + HO$_2$ → HIO$_3$ + O$_2$                                                  (R17)

is very exothermic with a deep submerged barrier (Figure 12). A Rice-Ramsperger-Kassel-Markus (RRKM) calculation using

375     the MESMER program (Glowacki et al., 2012) with the molecular parameters in Table 1 indicates that the rate constant of R17 is $k$(290 K) ~ 3 × 10$^{-11}$ cm$^3$ molecule$^{-1}$ s$^{-1}$. Given that [HO$_2$] is typically 3 × 10$^8$ molecule cm$^{-3}$ at midday (Stone et al., 2012), IO$_3$ could be converted to HIO$_3$ with a reaction rate of ~ 0.01 s$^{-1}$ i.e. about 15 times slower than the upper limit to the photolysis rate discussed above. R17 could therefore provide a route to HIO$_3$ formation, and this should be the subject of future investigation.

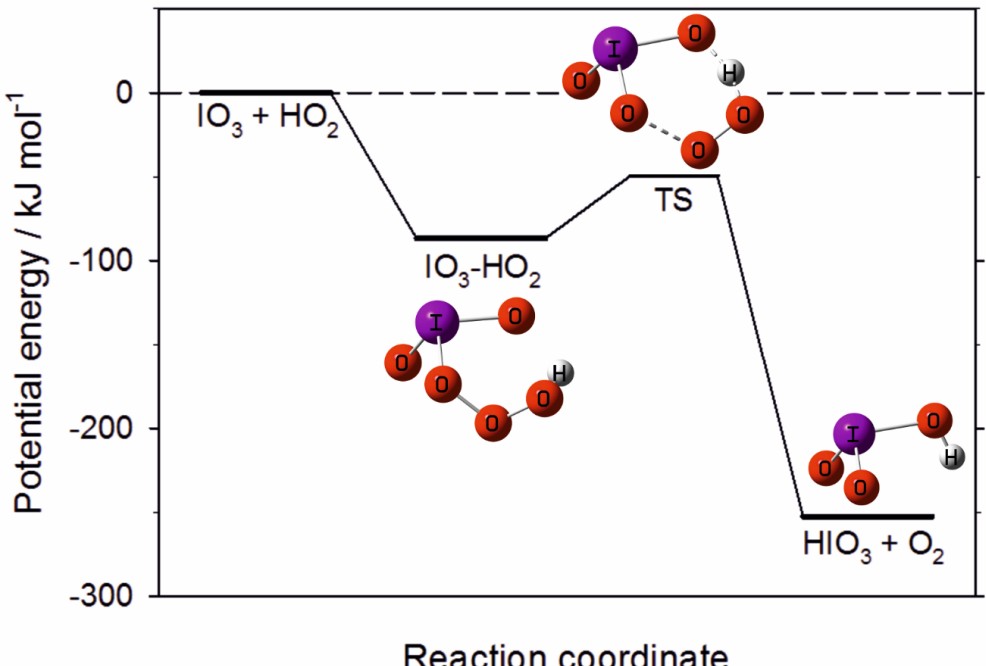

**Figure 12: Potential energy surface for the reaction between IO₃ and HO₂ to produce HIO₃, calculated at the B3LYP/gen level of theory (see text for further details).**

## 4.3. Atmospheric implications

The theoretical spectra constrained with the measurements at 355 nm and 532 nm have been implemented in CAM-Chem in order to calculate photolysis rates for $I_2O_y$ (y = 2, 3, 4) and $I_3O_7$. Having anchored the theoretical absorption spectra at 355 nm, we then assume that photolysis is possible up to the dissociation limit (which is the case for all the transitions of the $I_xO_y$ (x > 1) molecules). A conservative estimate is a factor of 2 uncertainty in the $J$ values for these molecules. The main changes with respect to SL2014 are:

1. The new $I_2O_2$ cross sections are lower in the UV than those of the absorber Y used in SL2014 (Figure 9) and extend over a wider range of the actinic range.

2. $I_2O_3$ and $I_2O_4$ have respectively lower and higher absorption cross sections than in SL2014, as a result of replacing the absorption Z and the $I_2O_4$ spectrum in solution in favour of theoretical spectra constrained with experimental values. The cross section of $I_2O_3$ is lower, but only by a factor of ~2 in the important region between 350 nm and 450 nm. The cross sections of $I_2O_4$ increase by 1 order of magnitude with respect to the spectrum in solution.

3. Higher iodine oxides like $I_3O_7$ and $I_5O_{12}$ not included in SL2014 also photolyze. They show bands in the green part of the spectrum (Figure 10) and therefore have higher photolysis rates than $I_2O_4$.

Annual zonal-average photolysis rate plots are shown in Figure 13 and Figure 14. Surface atmospheric photolysis of all the $I_xO_y$ species is generally fastest between 30° and -30° latitude, displaying slight minima at 5° and -5° latitude due to attenuation of actinic flux by cloud cover at the intertropical convergence zone. Average $I_xO_y$ global $J$ values in the upper troposphere exhibit a maximum around 5° latitude, where solar flux is highest. At 5° latitude, $I_2O_2$ and $I_2O_4$ $J$ values are largest at an altitude of 8 km. Similarly, the $J$ values for $I_2O_3$ and $I_3O_7$ at 5° latitude are highest at 8 km, and then remain relatively constant above 20 km. In contrast with $J$ values for $I_xO_y$ at 5° latitude, values at mid-latitudes increase with altitude, showing no maxima at 8 km. The observed increase in $J$ values at 8 km is explained by the albedo effect of cloud cover in certain regions, particularly the intertropical convergence zone, contributing to the total (incident + reflected) solar flux in these regions.

Figures 13(d-f) show the percentage difference between the SL2014 $J$ values and those calculated in this study. As a result of the new $I_2O_2$ cross sections extending across the actinic window, this species shows an increase in $J$ values at all latitudes, with the largest increases seen at the poles (~300% increase), and lower increases of around 100% near the equator. $I_2O_3$ shows a uniform decrease of around 46% across all latitudes and altitudes, as expected from the approximately uniform decrease of the new spectrum with respect to absorber Z, which was taken as $I_2O_3$ in SL2014. $I_2O_4$ exhibits the largest increase in $J$ values across all latitudes, ranging from 750% at 40 km between -30° and 30° latitude, up to increases above 1000% at the poles.

Photolytic lifetimes of $I_xO_y$ species in the troposphere vary from a few seconds for the most photolabile species, i.e. $I_3O_7$, to several minutes for $I_2O_3$, which has the smallest absorption cross-sections in the actinic region and the smallest $J$ values of the species in this study. Mid and high-latitudes exhibit particularly low $J$ surface values and encompass many coastal regions where active iodine concentrations are high and iodine-driven particle formation has been reported (Saiz-Lopez et al., 2012b).

Photolysis of higher iodine oxides (R9-R11) competes with gas-to-particle formation (R7) and recycles $IO_x$, which participates in ozone depleting cycles (e.g. R1-R3). Therefore, higher $J$ values at the surface tend to hinder particle formation and enhance ozone depletion. The new spectra result in decreased photolysis lifetimes of $I_2O_2$ (factor of ~3) and $I_2O_4$ (factor of ~10) and indicate that photolysis of nascent iodine clusters such as $I_3O_7$ and $I_5O_{12}$ may further slow down particle formation. On the other hand, the lifetime $I_2O_3$ is approximately doubled, which increases the chances of participating in other reactions, including clustering with other $I_xO_y$. Whether this may balance the decrease in the photolysis lifetimes of other species or not

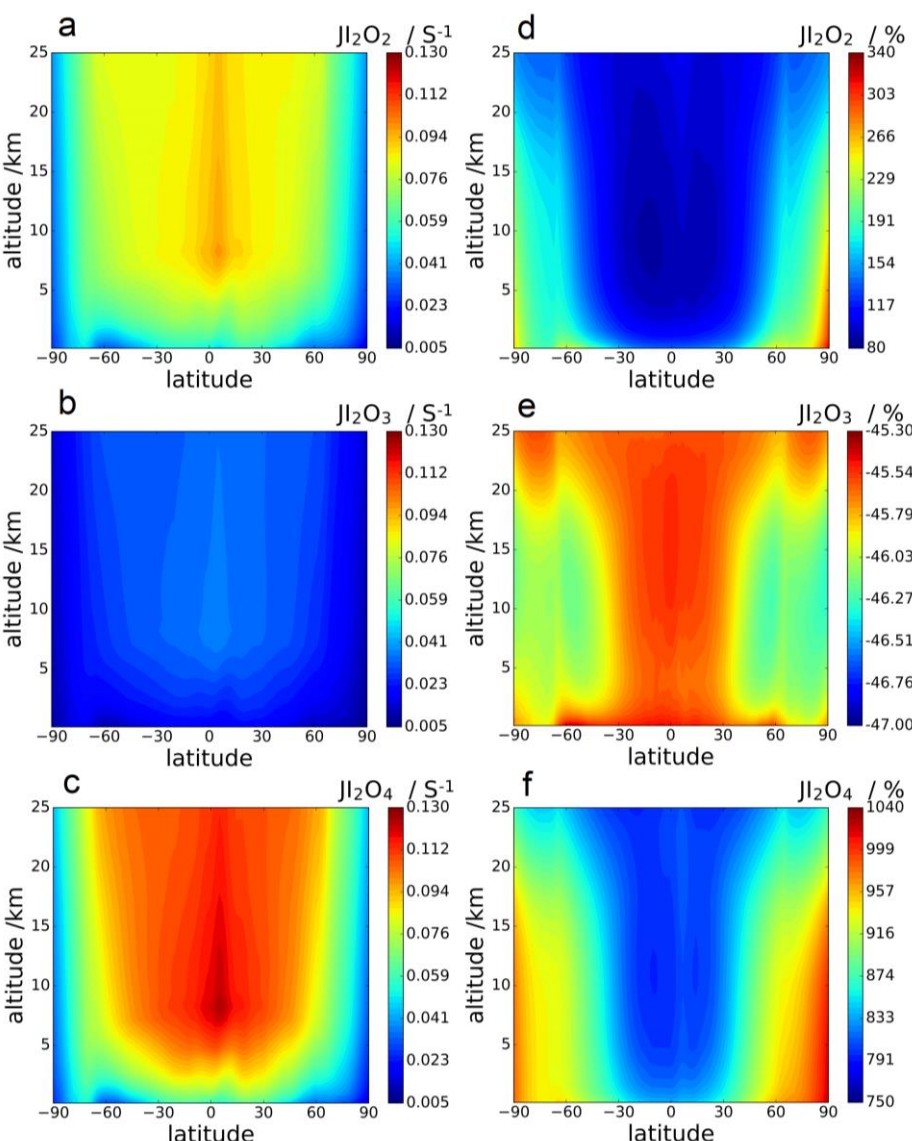

**Figure 13: Annual zonal average J value vertical profiles for $I_2O_2$, $I_2O_3$ and $I_2O_4$ (panels a, b and c, respectively; note that the colour scale is the same for these three panels) and the percentage change between the SL2014 annual zonal average J value vertical profiles, and the profiles in this study (panels d, e and f, respectively)**

remains to be studied in future work. $I_2O_3$ is believed to be an important iodine reservoir and key intermediate in the formation

of new particles (Gómez Martín et al., 2013; Saiz-Lopez et al., 2014), but there is a substantial lack of mechanistic information about iodine oxides and how they may convert (or not) into $HIO_3$.

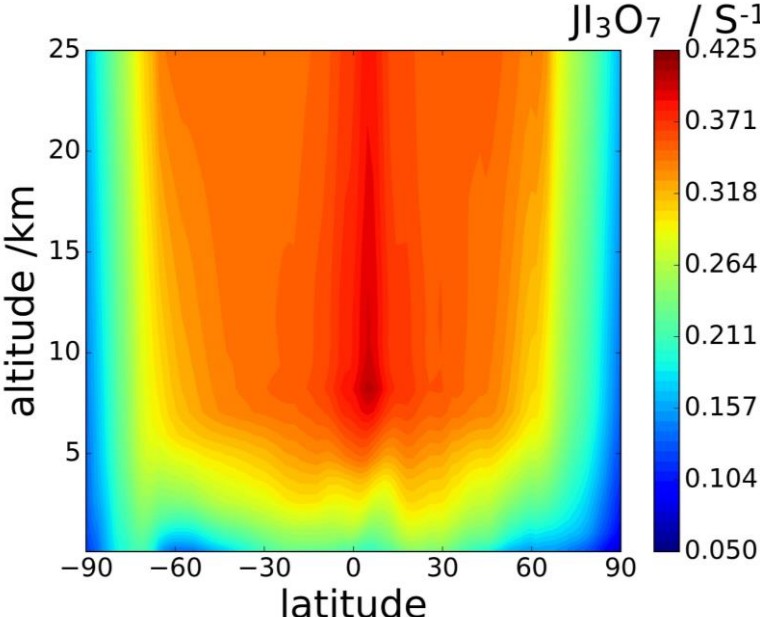

**Figure 14: Annual zonal average vertical J value profile for $I_3O_7$ (not included in SL2014)**

Another active iodine chemistry region is the tropical upper troposphere - lower stratosphere region (UTLS), where significant concentrations of iodine exist (Koenig et al., 2020) as a result of strong convection carrying iodine precursors emitted at the ocean surface (mainly $CH_3I$). The effect of the new spectra on the $J$ values in this region is similar to the effect at the surface,

although the changes are less pronounced. In the UTLS, $I_xO_y$ molecules play the role of iodine reservoirs and, as a result of the enhanced actinic flux (see $J$ maxima between 8 and 17 km around the equator in Figures 12 and 13), their photolysis has a significant impact on ozone depletion in this climatically sensitive region (Saiz-Lopez et al., 2012a). Iodine recycling in aerosol is believed to play an important role in explaining observed concentrations of gas-phase $IO_x$ (Koenig et al., 2020). The form in which gas-phase iodine (oxides, oxyacids or nitrate) is taken up in sulphate aerosol may also determine the extent to which

iodine can return to the gas phase.

To investigate with more detail the effect of the new spectra on particle formation, iodine speciation in aerosol and ozone depletion, the iodine chemistry scheme in CAM-Chem needs to be expanded to include the photolysis products and yields of $I_xO_y$ photolysis reactions, as well as the reactions of the possible photoproducts, which are poorly known. New laboratory and theoretical work is required to solve these uncertainties. The mechanism of IOP formation remains controversial, with two

possible pathways proposed via $I_xO_y$ and $HIO_3$. Work on the IOP formation mechanism conducted in the course of this research will be published elsewhere. Future laboratory measurements of $I_xO_y$ spectra and kinetics may benefit from a multiplexed

approach where tuneable photoionization mass spectrometry, absorption spectroscopy and multi-wavelength laser photodepletion are used concurrently. A follow up modelling study will be conducted with an updated mechanism for atmospheric iodine chemistry, including the photochemistry outlined in this paper, as well as a chemical scheme for IOP

formation.

## 5. Conclusion

The photodepletion laboratory experiments reported in this work confirm that $I_xO_y$ species are photolabile, and therefore supports the $J(I_xO_y)$ scenario in ASL2014, with important consequences for ozone depletion. The values of the absorption cross sections obtained at 355 nm and 532 nm have been employed to revisit the assignment of absorption bands in previous

work. Also, comparison between theoretical spectra and measurements gives some clues about the geometry of the molecules. $I_2O_3$ has a smaller absorption cross-section in the actinic region than previously thought, which may facilitate other reactions, including clustering to form new particles. Other $I_xO_y$ species have larger absorption cross-sections in the actinic region, resulting in rapid photolysis to smaller iodine-containing molecules. Possible photolysis products include $IO_3$, which could be a precursor of $HIO_3$ in the marine boundary layer. These new findings highlight the need for new experimental and theoretical

studies, particularly investigating the products of these photochemical processes, as well as the effect on IOPs (and therefore cloud condensation nuclei) formation.

**Author contribution.** A. S.-L. devised the research. J. C. G. M., M. A. B. and J. M. C. P. designed the experimental set up, and T. R. L. carried out the experiments and analysed the data; J. M. C. P. and J. C. G. M. carried out electronic structure and master equation calculations; C. A. C. carried out radiative transfer calculations; T. R. L. and J. C. G. M. wrote the manuscript

with contributions from all co-authors.

**Code/Data availability**. The data that support the findings of this study are available from the corresponding authors on reasonable request.

**Competing interests.** The authors declare that they have no conflict of interest.

**Acknowledgements.** This study has received funding from the European Research Council Executive Agency under the

European Union's Horizon 2020 Research and Innovation program (Project 'ERC-2016-COG 726349 CLIMAHAL'). J. C. G. M. acknowledges financial support from the State Agency for Research of the Spanish MCIU through the "Center of Excellence Severo Ochoa" award to the Instituto de Astrofísica de Andalucía (SEV-2017-0709) and the Ramon y Cajal program (RYC-2016-19570). J. C. G. M. and A. S.-L. acknowledge support of the publication fee by the CSIC Open Access Publication Support Initiative through its Unit of Information Resources for Research (URICI).

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

# APPENDIX A. Molecular geometries and parameters of iodine oxides at B3LYP/6-311+G(2d,p) (+AE) level of theory and benchmarking of the TD-DFT method for iodine oxides

 Table A1: Molecular properties of $I_2O_3$, $I_2O_4$, $I_3O_6$ (cyclic), $I_3O_6$ (linear), $I_3O_7$ (cyclic), $I_3O_7$ (linear), $IO_3$ and the transition state for dissociation of $IO_3$ to $IO + O_2$ (illustrated in Figure A1); and the stationary points on the $IO_3 + HO_2$ surface (Figure 12).

| Molecule | Cartesian co-ordinates /Å [a] | Rotational constants /GHz [a] | Vibrational frequencies /cm$^{-1}$ [a] |
|---|---|---|---|
| $I_2O_3$ | I, -0.009, -0.201, -0.192<br>O, -1.208, -0.160, 1.377<br>I, -0.583, 0.643, 3.128<br>O, 0.427, 1.521, -0.391<br>O, 1.423, -1.065, 0.438 | 4.46970, 0.57042, 0.55739 | 44, 88, 241, 256, 299, 450, 624, 872, 902 |
| $I_2O_4$ | O, -1.655, -0.203, 1.563<br>I, 0.046, 0.313, 1.770<br>O, 0.699, 0.048, -0.103<br>O, 0.936, -0.982, 2.619<br>I, -0.385, -0.913, -1.502<br>O, -1.272, 0.428, -2.370 | 3.55600, 0.49098, 0.47135 | 48, 67, 92, 164, 248, 260, 308, 444, 538, 812, 874, 906 |
| $I_3O_6$ (cyclic) | I, 1.335, -1.791, -0.031<br>O, 1.569, 0.091, -0.823<br>O, 3.064, -2.228, 0.378<br>I, 1.319, 1.750      , 0.186<br>O, -0.279, 1.411, 0.952<br>O, 1.051, 2.850, -1.192<br>I, -2.620, -0.026, 0.066<br>O, -1.412, -1.310, -0.437<br>O, -4.280, -0.729, -0.202 | 0.44642, 0.26455, 0.17279 | 18, 26, 54,59, 66, 70, 87, 107, 116, 171, 202, 274, 295, 319, 444, 563, 783, 804, 814, 843, 898 |
| $I_3O_6$ (linear) | O, -0.860, 0.445, 0.841<br>I, -0.142, -0.977, -0.191<br>O, 1.698, -0.828, 0.694<br>O, -2.369, -1.079, -0.706<br>I, 3.054, 0.513, 0.198<br>O, 2.321, 1.165, -1.301<br>O, 4.415, -0.561, -0.236<br>I, -3.219, 0.313, 0.228<br>O, -3.185, 1.803, -0.797 | 1.40546, 0.15363, 0.14491 | 21, 33, 60, 69, 85, 98, 122, 166, 184, 262, 268, 302, 313, 343, 442, 565, 664, 699, 843, 871, 903 |
| $I_3O_7$ (cyclic) | I, 1.050, -1.602, -0.490<br>O, 1.254, 1.127, -0.500<br>O, 1.159, -3.334, -0.902<br>I, 0.374, 2.634, 0.070<br>O, -1.810, 0.925, 0.501<br>O, 1.558, 4.011, -0.093<br>I, -2.237, -0.789, 0.151<br>O, 0.594, -1.505, 1.245<br>O, -3.595, -0.762, -1.001<br>O, -0.813, -1.235, -1.152 | 0.43718, 0.25525, 0.17036 | 29, 49, 61, 66, 79, 82, 90, 111, 123, 171, 186, 262, 279, 302, 308, 324, 494, 566, 782, 807, 852, 862, 905, 909 |

| | | | |
|---|---|---|---|
| I$_3$O$_7$ (linear) | O, 4.334, 1.318, 0.098<br>I, 3.552, -0.283, 0.217<br>O, 1.828, 0.194, 0.994<br>O, 2.970, -0.766, -1.411<br>I, 0.004, 0.013, 0.002<br>O, -1.823, -0.085, -0.997<br>O, 0.008, -1.794, 0.247<br>I, -3.543, -0.351, -0.118<br>O, -2.958, -0.382, 1.579<br>O, -4.337, 1.219, -0.426 | 2.11361, 0.11760, 0.11718 | 21, 25, 33, 44, 46, 60, 84, 140, 186, 203, 263, 266, 296, 312, 320, 418, 433, 552, 635, 759, 867, 867, 901, 902 |
| IO$_3$ | I, 0.080, 1.235, 0.000<br>O, 0.645, -0.481, 0.000<br>O, 0.645, 2.092, 1.486<br>O, 0.645, 2.092, -1.486 | 6.18204, 6.18204, 3.57497 | 258, 258, 268, 766, 766, 799 |
| TS for IO$_3$ → IO + O$_2$ | I, 0.205, 0.253, -0.082<br>O, 1.400, -1.046, 0.249<br>O, -1.406, -0.589, -0.790<br>O, -1.419, -0.181, 0.909 | 8.93231, 4.04781, 3.81028 | -914, 142, 252, 354, 701, 867 |
| IO$_3$-HO$_2$ complex | I, -0.649, 0.251, 0.072<br>O, -0.420, -1.455, 0.587<br>O, -0.429, 1.299, 1.501<br>O, 1.098, 0.455, -0.961<br>O, 2.104, -0.347, -0.326<br>O, 2.090, -1.626, -0.927<br>H, 1.449, -2.095, -0.357 | 4.44296, 1.49686, 1.34348 | 64, 143, 192, 226, 297, 316, 464, 530, 568, 719, 860, 903, 938, 1453, 3638 |
| TS from IO$_3$-complex to HIO$_3$ + O$_2$ | I, 0.131, 1.178, -0.047<br>O, 0.550, -0.594, 0.341<br>O, 0.383, 2.058, 1.482<br>O, 1.579, 1.575, -1.172<br>O, 2.931, 0.475, -0.440<br>O, 2.661, -0.726, -0.774<br>H, 1.671, -0.878, -0.163 | 4.24271, 1.59293, 1.46409 | 1000$i$, 74, 225, 250, 288, 323, 377, 500, 624, 711, 823, 904, 1246, 1306, 1749 |
| HIO$_3$ | I, 1.574, -0.480, 0.151<br>O, -0.137, -0.710, -0.315<br>O, 2.309, 0.401, -1.439<br>O, 1.647, 0.832, 1.363<br>H, 1.792, 1.203, -1.629 | 5.68293, 5.36595, 3.62995 | 62, 258, 274, 304, 584, 871, 901, 981, 3732 |

[a] Calculated at the B3LYP/gen level of theory (see text)


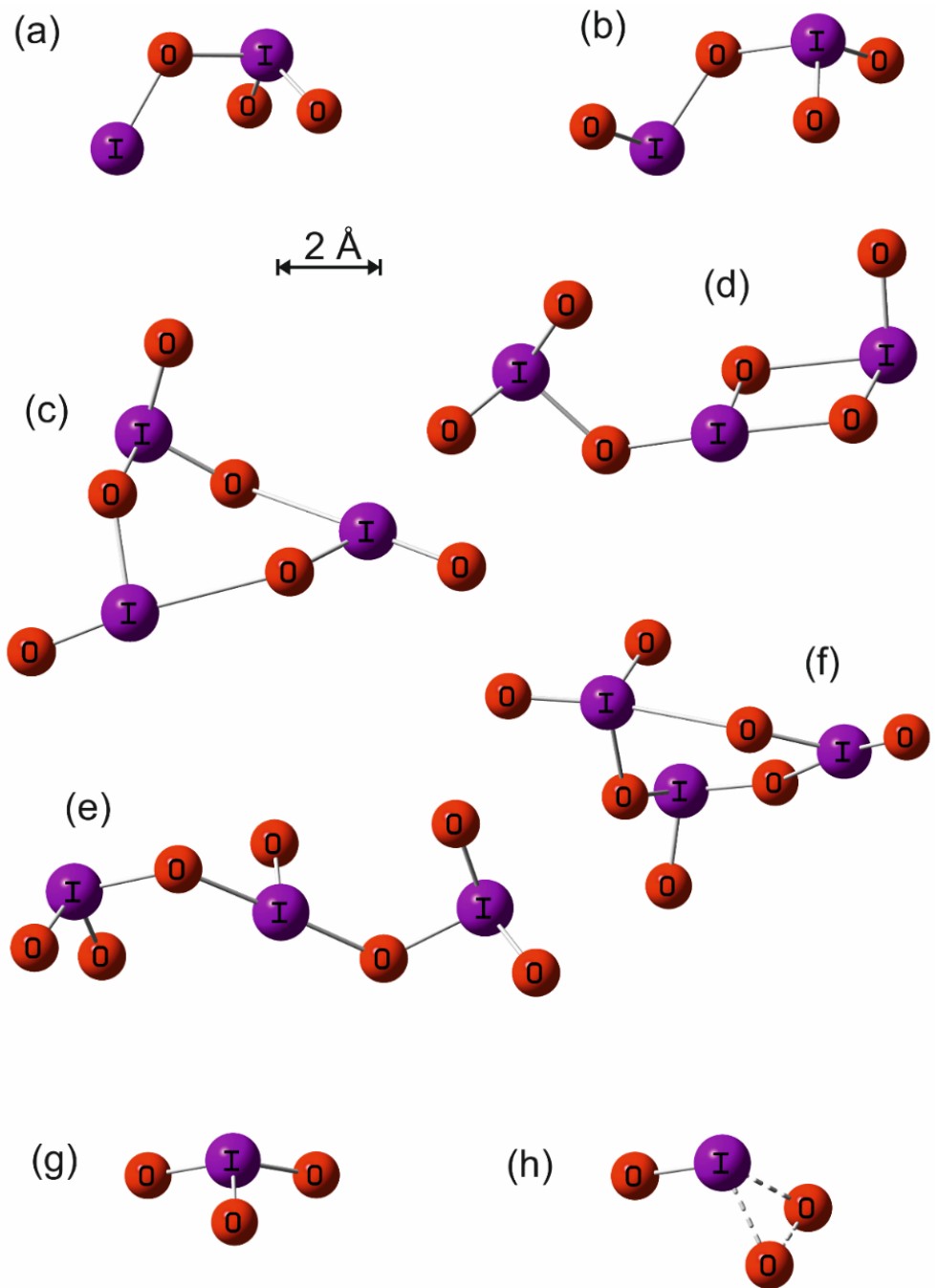

**Figure A1: Molecular geometries of $I_2O_3$, $I_2O_4$, $I_3O_6$ (cyclic), $I_3O_6$ (linear), $I_3O_7$ (cyclic), $I_3O_7$ (linear), $IO_3$ and the transition state for dissociation of $IO_3$ to $IO + O_2$, calculated at the B3LYP/gen level of theory (see main text for details).**

 **Table A2.** Peak wavelengths and oscillators strengths ($f$) for the first 30 excited states of HOI, IO, OIO and $IO_3$ at the TD/B3LYP/G2 level of theory.

| HOI | | IO | | OIO | | $IO_3$ | |
|---|---|---|---|---|---|---|---|
| $\lambda$ / nm | $f$ | $\lambda$ / nm | $f$ | $\lambda$ / nm | $f$ | $\lambda$ / nm | $f$ |
| 450.3 | 0.0001 | 94003.7 | 0 | 598.9 | 0 | 1378.3 | 0 |
| 367.0 | 0.0027 | 513.0 | 0 | 586.8 | 0.0002 | 951.7 | 0.0039 |
| 224.7 | 0.0089 | 390.6 | 0 | 584.2 | 0.0217 | 951.0 | 0.0039 |
| 192.8 | 0.0478 | 386.7 | 0.0001 | 342.6 | 0.0006 | 719.7 | 0.0081 |
| 178.4 | 0.0999 | 379.4 | 0.0365 | 337.5 | 0 | 719.3 | 0.0081 |
| 173.3 | 0.072 | 338.5 | 0.0001 | 281.0 | 0 | 264.4 | 0.0027 |
| 168.9 | 0.1978 | 321.0 | 0.0001 | 278.7 | 0 | 264.4 | 0.0027 |
| 165.4 | 0.0447 | 244.5 | 0.0052 | 270.3 | 0.0027 | 254.9 | 0.0127 |
| 164.9 | 0.002 | 230.9 | 0.0001 | 262.6 | 0 | 249.4 | 0 |
| 161.4 | 0.0659 | 212.4 | 0.0006 | 262.0 | 0 | 246.4 | 0.0001 |
| 157.9 | 0.0003 | 198.7 | 0.0114 | 249.7 | 0.0046 | 246.3 | 0.0001 |
| 153.2 | 0.0056 | 193.5 | 0.0249 | 247.5 | 0.0002 | 237.2 | 0.011 |
| 139.2 | 0.082 | 185.3 | 0.085 | 232.7 | 0.0009 | 222.1 | 0.001 |
| 133.4 | 0.0191 | 179.6 | 0.0112 | 231.1 | 0.0008 | 222.0 | 0.001 |
| 131.4 | 0.1567 | 173.2 | 0 | 229.9 | 0 | 221.5 | 0 |
| 130.7 | 0.0911 | 172.9 | 0.009 | 228.1 | 0.0459 | 213.8 | 0.0029 |
| 121.7 | 0.0214 | 167.4 | 0 | 219.8 | 0.0002 | 213.7 | 0.0028 |
| 121.4 | 0.071 | 167.3 | 0.0642 | 206.5 | 0 | 205.0 | 0 |
| 118.8 | 0.0341 | 165.8 | 0.02 | 200.6 | 0.0063 | 205.0 | 0 |
| 118.6 | 0.0023 | 163.6 | 0.0007 | 188.6 | 0.0279 | 197.4 | 0.0013 |
| 117.2 | 0.0009 | 157.9 | 0.0005 | 173.8 | 0.1771 | 195.2 | 0.0019 |
| 116.0 | 0.0059 | 152.0 | 0.037 | 167.5 | 0.0123 | 195.2 | 0.0019 |
| 114.9 | 0.067 | 150.7 | 0 | 167.2 | 0.0064 | 193.9 | 0.0076 |
| 112.3 | 0.0094 | 143.6 | 0 | 166.4 | 0.0007 | 193.8 | 0.0075 |
| 112.0 | 0 | 142.0 | 0.0836 | 165.4 | 0 | 192.3 | 0 |
| 111.0 | 0.037 | 140.5 | 0.4635 | 164.6 | 0 | 190.2 | 0.0085 |
| 110.3 | 0.0068 | 131.0 | 0 | 162.0 | 0.0002 | 190.2 | 0.0083 |
| 109.3 | 0.073 | 130.9 | 0.0107 | 155.5 | 0 | 187.1 | 0.015 |
| 105.8 | 0.0404 | 130.1 | 0.0358 | 155.1 | 0.0004 | 186.3 | 0 |
| 105.4 | 0.4533 | 126.0 | 0.0663 | 155.0 | 0.0017 | 185.0 | 0.0145 |

**Table A3.** Peak wavelengths and oscillators strengths ($f$) for the first 30 excited states of $I_2O_2$, $I_2O_3$, and $I_2O_4$ at the TD/B3LYP/G2 level of theory.

| $I_2O_2$ | | $I_2O_3$ | | $I_2O_4$ | |
|---|---|---|---|---|---|
| $\lambda$ / nm | $f$ | $\lambda$ / nm | $f$ | $\lambda$ / nm | $f$ |
| 758.1 | 0.0063 | 518.9 | 0 | 676.5 | 0.0051 |
| 563.0 | 0 | 455.5 | 0.0045 | 366.0 | 0.0039 |
| 480.6 | 0.0062 | 335.2 | 0.034 | 344.7 | 0.0008 |
| 367.6 | 0.0052 | 334.6 | 0.0006 | 323.6 | 0.0348 |
| 338.9 | 0.0108 | 297.5 | 0.0046 | 319.4 | 0.0375 |
| 306.4 | 0.0478 | 286.1 | 0.0025 | 291.7 | 0.0342 |
| 300.0 | 0.0009 | 265.6 | 0.0061 | 287.3 | 0.0017 |
| 275.3 | 0.0358 | 258.6 | 0.0419 | 280.1 | 0.079 |
| 270.3 | 0.0047 | 240.7 | 0.0532 | 267.9 | 0.0085 |
| 257.1 | 0.0163 | 236.0 | 0.0009 | 253.3 | 0.0124 |
| 255.4 | 0.0017 | 229.0 | 0.0068 | 248.1 | 0.0082 |
| 252.7 | 0.1031 | 227.5 | 0 | 241.0 | 0.0058 |
| 226.1 | 0.0106 | 225.7 | 0.002 | 237.0 | 0.0116 |
| 219.1 | 0.0049 | 223.6 | 0.0002 | 231.6 | 0.0059 |
| 200.2 | 0.0081 | 217.4 | 0.0005 | 228.1 | 0.0086 |
| 199.3 | 0.0406 | 215.7 | 0.0129 | 225.9 | 0.001 |
| 192.8 | 0.0187 | 214.7 | 0.0169 | 218.9 | 0.0145 |
| 186.8 | 0.0034 | 211.4 | 0.0039 | 216.4 | 0.0101 |
| 184.6 | 0.0041 | 209.0 | 0.0332 | 213.8 | 0.0181 |
| 184.1 | 0.009 | 200.7 | 0.0183 | 209.6 | 0.0124 |
| 179.0 | 0.0667 | 199.1 | 0.0019 | 207.6 | 0.002 |
| 177.8 | 0.0844 | 197.1 | 0.1393 | 206.8 | 0.0026 |
| 177.4 | 0.0251 | 190.5 | 0.0454 | 202.3 | 0.0031 |
| 172.6 | 0.0911 | 185.7 | 0.0031 | 199.1 | 0.0044 |
| 170.0 | 0.0251 | 175.2 | 0.0321 | 197.3 | 0.0016 |
| 168.5 | 0.007 | 174.6 | 0.0031 | 196.3 | 0.0101 |
| 165.2 | 0.0301 | 170.2 | 0.1018 | 194.1 | 0.0283 |
| 162.7 | 0.0146 | 169.6 | 0.0256 | 193.7 | 0.0071 |
| 162.1 | 0.0058 | 168.9 | 0.062 | 191.9 | 0.0036 |
| 161.2 | 0.091 | 167.3 | 0 | 190.3 | 0.0036 |

**Table A4.** Peak wavelengths and oscillators strengths (*f*) for the first 30 excited states of $I_3O_6$ linear, $I_3O_6$ cyclic, $I_3O_7$ linear and $I_3O_7$ cyclic at the TD/B3LYP/G2 level of theory.

| $I_3O_6$ linear | | $I_3O_6$ cyclic | | $I_3O_7$ linear | | $I_3O_7$ cyclic | |
|---|---|---|---|---|---|---|---|
| $\lambda$ / nm | *f* | $\lambda$ / nm | *f* | $\lambda$ / nm | *f* | $\lambda$ / nm | *f* |
| 1010.2 | 0.0052 | 1235.0 | 0 | 880.9 | 0.0002 | 643.1 | 0.0001 |
| 554.6 | 0.0095 | 850.6 | 0.0067 | 782.1 | 0.0005 | 546.0 | 0.0287 |
| 486.0 | 0.0242 | 771.7 | 0.0023 | 657.6 | 0.0001 | 517.8 | 0.0002 |
| 467.9 | 0.0005 | 630.7 | 0.0001 | 606.3 | 0.0001 | 434.1 | 0.0022 |
| 452.6 | 0.0123 | 550.4 | 0.0229 | 601.9 | 0.0007 | 427.1 | 0.0003 |
| 432.5 | 0.0002 | 531.2 | 0.0001 | 542.8 | 0.002 | 389.5 | 0.0014 |
| 424.1 | 0.0032 | 518.5 | 0 | 529.3 | 0.0006 | 386.1 | 0.0002 |
| 416.0 | 0.014 | 462.3 | 0.0001 | 490.3 | 0.0089 | 364.3 | 0.0004 |
| 403.2 | 0.0047 | 433.1 | 0 | 479.5 | 0.005 | 357.1 | 0.0002 |
| 390.8 | 0.0033 | 392.7 | 0.0004 | 462.5 | 0.0089 | 355.6 | 0.0001 |
| 384.2 | 0.004 | 387.8 | 0.0082 | 382.9 | 0.0054 | 351.9 | 0.002 |
| 379.0 | 0.0031 | 371.9 | 0.0002 | 376.9 | 0.0006 | 347.7 | 0.0005 |
| 361.9 | 0.0203 | 363.2 | 0 | 364.0 | 0.001 | 343.5 | 0.012 |
| 353.4 | 0.0154 | 354.1 | 0.0008 | 353.5 | 0 | 338.7 | 0.0042 |
| 347.0 | 0.0009 | 351.2 | 0.0013 | 340.1 | 0.0052 | 334.2 | 0.0039 |
| 344.8 | 0.0159 | 349.1 | 0.0002 | 333.9 | 0.0004 | 331.7 | 0.0108 |
| 336.5 | 0.0117 | 339.8 | 0.0023 | 331.7 | 0.0032 | 329.8 | 0.0001 |
| 329.5 | 0.0073 | 333.3 | 0.0001 | 311.0 | 0 | 326.3 | 0.001 |
| 323.7 | 0.0024 | 332.1 | 0 | 305.6 | 0.082 | 322.0 | 0.0022 |
| 321.8 | 0.0044 | 325.4 | 0.0026 | 304.5 | 0.0737 | 320.2 | 0.0026 |
| 317.7 | 0.0015 | 325.0 | 0.0009 | 300.9 | 0.0001 | 314.1 | 0.0019 |
| 316.2 | 0.0079 | 323.1 | 0.0017 | 295.3 | 0.0005 | 310.1 | 0.0031 |
| 301.6 | 0.0035 | 318.5 | 0.0081 | 290.8 | 0.0013 | 302.7 | 0.0004 |
| 297.6 | 0.0115 | 315.8 | 0.008 | 290.2 | 0.0013 | 298.5 | 0.0003 |
| 295.8 | 0.002 | 311.9 | 0.0028 | 289.9 | 0.0028 | 297.4 | 0.0007 |
| 295.1 | 0.0234 | 309.0 | 0.0521 | 284.9 | 0.0055 | 293.3 | 0.0015 |
| 290.2 | 0.0001 | 308.0 | 0.0011 | 279.1 | 0.0005 | 292.6 | 0.0004 |
| 284.7 | 0.0015 | 306.2 | 0.0015 | 275.3 | 0 | 289.0 | 0.004 |
| 282.3 | 0.0102 | 302.8 | 0.0014 | 274.8 | 0.0018 | 285.4 | 0.0034 |
| 279.0 | 0.0059 | 299.0 | 0.0014 | 269.4 | 0.0003 | 282.2 | 0.0025 |



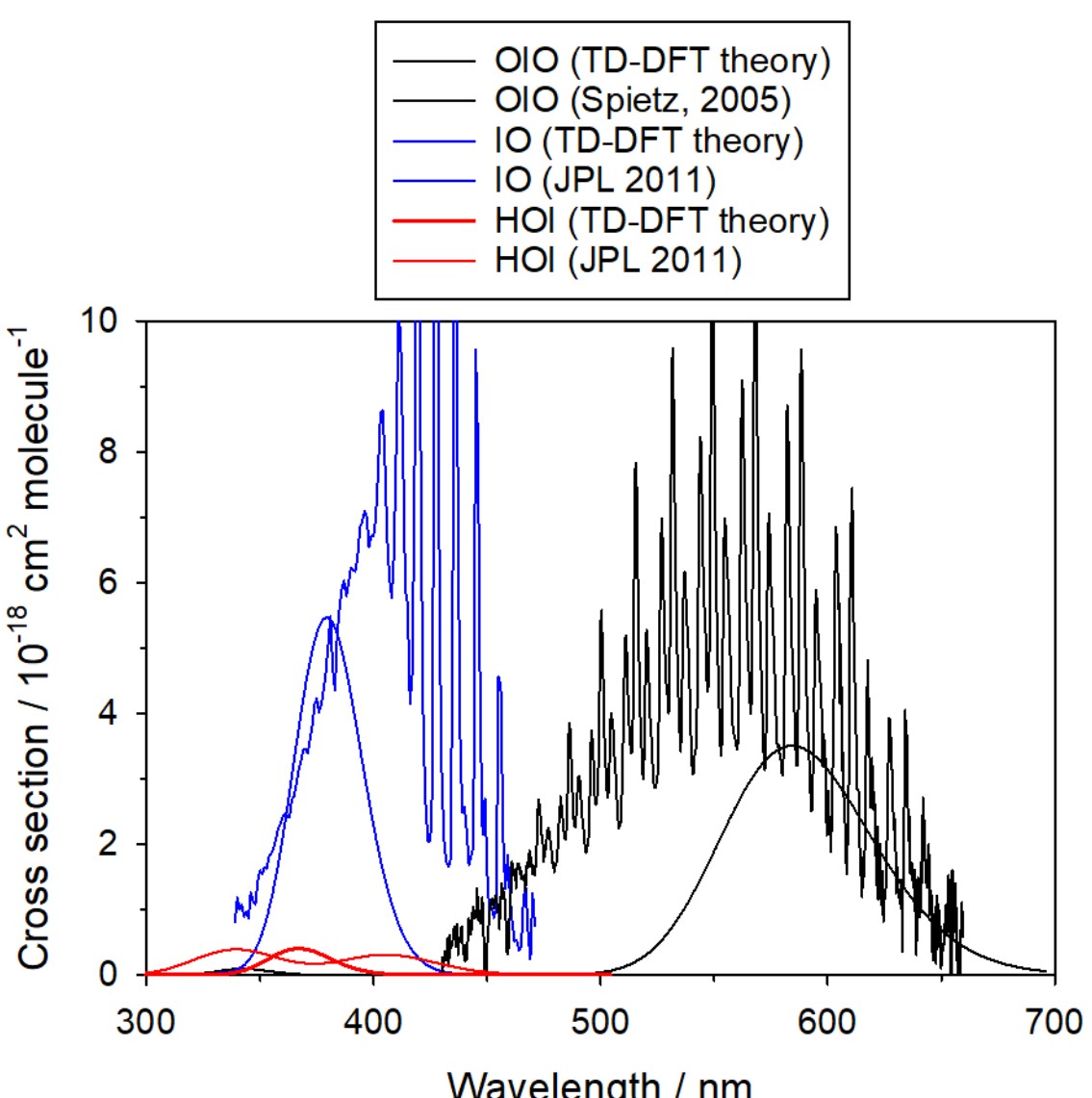

**Figure A2: Experimental absorption cross sections of IO (Sander et al., 2011), OIO (Spietz et al., 2005) and HOI (Sander et al., 2011) and absorption spectra calculated with the TD-DFT method (this work).**


## APENDIX B. Photofragmentation of I$_x$O$_y$ species

The photofragmentation of I$_x$O$_y$ species to product ions in the photoionization chamber of the detection apparatus necessitates careful experimentation in order to ensure that any photodepletion of a species of interest is solely due to its 355 nm or 532 nm photolysis in the flow tube, and not obscured by the product ions of larger I$_x$O$_y$ species. To elucidate the optimal window for investigating photodepletion of each species, time resolved mass-spectra were recorded for each set of experimental conditions (Figure 5). From the kinetic information, it is then possible to inspect the averages of the mass spectra within these windows to ensure that the species of interest is present, but larger I$_x$O$_y$ species are not, as shown in Figure B1. By ensuring no larger I$_x$O$_y$ species are present, it follows that for a species of interest, only precursor ions of the species are contributing to the signal intensity within that time window, and that no contribution to the recorded signal is coming from product ions. Note that the signals shown in Figure 5 and Figure B1 are not accumulated for extended periods of time, and as such are relatively noisy. Long accumulation times and corresponding large signal to noise ratios are unnecessary for these experiments, since the objective is simply to elucidate optimal time-delays within which photodepletion experiments are carried out, (photodepletion experiments are typically carried for ~10× as many accumulations) and are carried out prior to an experimental session. It should be noted also that the optimal timescales such as those shown in Figure 5 and Figure B1, vary depending on the concentration of IO formed at the beginning of the reaction sequence, and since the reactions which facilitate the stepwise formation of the higher oxides are second-order, even modest changes in [IO] at early times can result in significant changes to the appearance times of the different species of interest.

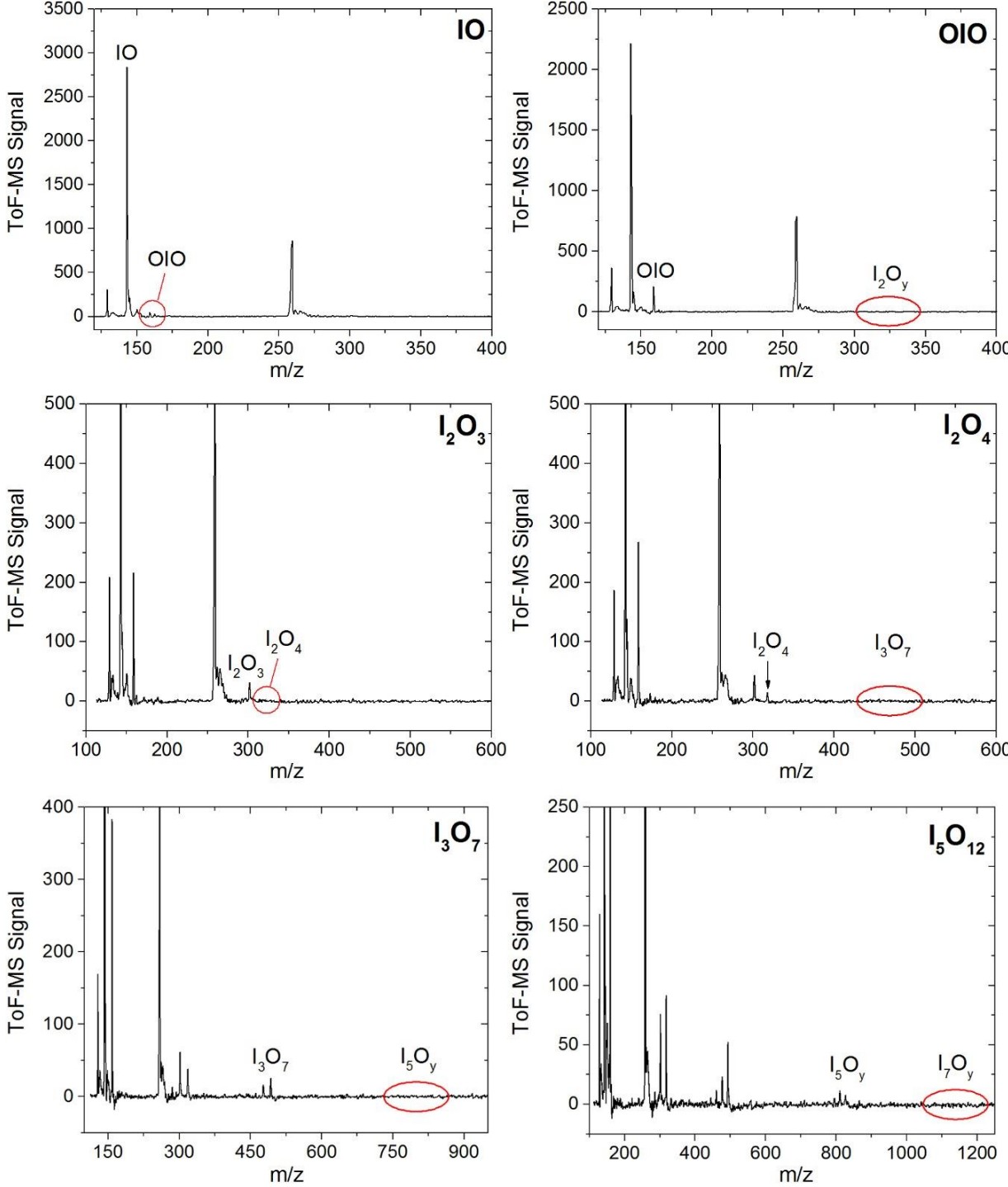

**Figure B1: Mass spectra corresponding to the optimised time delays shown in Figure 5. The plots are generated by averaging the signal obtained for each mass over the 5 ms window. The species of interest is given in the top right corner of each spectrum.**