# Peer review of "Determination of the absorption cross-sections of higher order iodine oxides at 355 nm and 532 nm."

_Atmospheric Chemistry and Physics, 2020_

## Referee Comment (RC1) · Anonymous Referee #1 · 17 Jun 2020

Lewis et al combine experimental and theoretical work examining the absorption cross-sections of higher iodine oxides (IxOy). The results indicate that most oxides are rapidly removed by photolysis, which has repercussions for formation of iodine containing particle. The experiments are complex as IxOy have to be generated in-situ, the data appears to be of reasonable quality. This review does not address the accuracy of the theoretical work. The manuscript is well written; the authors may consider the following suggestions for improvement.

Main comment:

Actinometry at 355 nm is using NO2 whereas OIO is used at 532 nm. The absorption

cross-section of OIO at 532 nm is quoted as being "relatively well known" (L174). The cross-sections of I2 are well known as are those of NO2. It is questionable whether an unstable trace-gas, the spectrum of which has only been measured following its transient formation as the product of the IO-self reaction can be termed "well known". The authors must be more quantitative here. They should properly review (and cite) the limited literature on the OIO cross-section at this wavelength. The same applies to the unity quantum yield of photolysis of OIO. Do all the literature studies agree on this value ? Again, the authors need to review (and cite) the literature and state why they believe the quantum yield is one, rather than simply citing one article in which one of the present authors also contributed.

L31 Presumably the 9-16% ozone depletion from iodine chemistry refers to the marine boundary layer. This should be made clear.

L77 "This is currently an important uncertainty..." What does "this" refer to? Perhaps "an important source of uncertainty" is better?

L114 Signal-to-noise ratio $\sim$ 400. S/N measured over what period of time? Is drift in laser intensity not more important than S/N for an absorption cell that has no reference photo-diodes. Limit of detection in OD units would be more useful.

Figure 4. What is the pulse-width of the excitation laser ? Please explain why the depletion of NO2 (presumably in $\sim$10 ns) takes 150 ms until the new plateau is reached. To which expression was the red curve fitted and why was it chosen?

L146 and L147 Laser "shot", excimer "flash". Perhaps use the term "pulse" ?

Figure 6 What causes the increase in the IO signal after 1.85 ms.

L259 Please show the trace obtained using the actinometer (OIO). This could potentially go into supplementary information along with the raw data (lack of signal change) for the higher oxides and would would give the reader an idea of the data-quality.

Figure 7 Indicate when the laser was triggered in the I2O3 experiment. Are the red

curves the same function as used in Figure 4 ?

L370 please provide references to the tropical UTLS being a "hotspot" of iodine chemistry. Have there been measurements of IO or higher iodine oxides in this part of the atmosphere?

---

## Referee Comment (RC2) · Anonymous Referee #2 · 22 Jun 2020

Review of "Determination of the absorption cross-sections of higher order iodine oxides at 355 nm and 532 nm" by Lewis et al.

This paper presents photochemical data for iodine oxides (IxOy), a hitherto elusive family of atmospheric transients thought to play an important role in the coastal marine boundary layer. Results were determined via laser photolysis / photodepletion experiments using mass-spectrometry detection of IxOy, supplemented by ab-initio calculations. This first extensive dataset can provide valuable information needed to construct models of atmospheric chemistry. However, there were a few points that require further clarification and discussion within the manuscript prior to publication in ACP.

[Figure]

Firstly, experiments appear to have been conducted under very different conditions of gas pressure and identity to those found in the boundary layer. At the top of page 6 it was stated that experimental pressures of 4 to 7 Torr were used (though whether of He or N2 was not clear). This is clearly a very different matrix of gases to 760 Torr of N2, O2 and H2O found in coastal boundary layer air. There is surely some doubt therefore, whether some qualitative observations from this work are valid for atmospheric models. The chemical mechanisms for IxOy formation will surely differ to some extent due to changes in stabilization rates for association products at the different gas-pressure, or perhaps reactions of transients with O2 (or even H2O) that would proceed faster in the boundary layer. Specifically, much is made of the absence of I4Oy species, but we simply do not know if these compounds would be formed in realistic atmospheric conditions. If there is clear evidence for why such pressure or O2 effects are unimportant then this needs to be detailed in the manuscript.

Second, regarding more quantitative results, could photolysis quantum yields and therefore photolysis cross-sections differ as the pressure and identity of the surrounding gas matrix changes? Certainly these effects can be important for many atmospheric transients, not least for the actinometer NO2 though at a longer wavelength than used in this work. I suspect that such pressure effects will have a negligible impact on the quantitative results from this work, given, as stated on page 16, the featureless nature of the spectra. However, given the large divergence from atmospheric conditions noted above, a strong statement to the effect that these cross-sections / quantum yields are applicable to realistic atmospheric conditions would be advisable.

Third regards the use of OIO as the actinometer for 532 nm experiments. I can understand why this molecule was used, given a limited set of alternatives. However, the manuscript needs to properly represent the problems that this introduces into the interpretation of results. The cross-section for OIO may be "relatively well known" as stated on page 8 (line 176), but I was not able to find a recent review justifying this statement. On the contrary, of the two references quoted in this work for OIO crosssections, only one (from Bloss et al.) quotes a value at 532 nm. A very quick look in the literature yielded cross-section data from five additional papers (Spietz 2005, Joseph 2005, Himmelmann 1996, Tucceri 2006 and from Ingham 2000) that differ by up to a factor of two at some wavelengths. Further, the quantum yield (QY) for atomic iodine production from OIO is certainly not established. Ingham et al. reported QY < 0.15 for this channel, in direct contrast to the value of unity taken from Gomez Martin et al. and used for the purposes of this work. Happily, a careful re-wording of the manuscript here can rectify these problems. First, the section on page 8 needs to take full account of the published literature on OIO. Second, the results obtained in this work need to be clearly denoted as being determined relative to the rather uncertain cross-sections and quantum yields for OIO at 532 nm.

Lastly, on Figure 4 "an empirical fit" was used to obtain depletion parameters. What was the function? More interestingly, which processes were responsible for the delay in signal depletion following (presumably rapid) photolysis. Is this delay instrumental, and therefore present in other experiments? It is not possible for the reader to assess for themselves, as a very different timescale is presented on Figs 6 and 7.

More minor comments: Page 2 it was stated that the main atmospheric fate of iodine atoms is reaction with O3 to form IO. This is likely true across much of the globe, but a significant alternative exists in polluted air (as encountered in many important areas of the coastal MBL) where reaction with NO2 to form INO2 would be competitive. The text on page 3 reports flows diluted in He whilst Figure 1 appears to indicate N2 as the principal diluent. Which is correct? If a mixture of the two then please use the text and / or the caption to Figure 1 to offer more detail. Similarly in the experimental details a laser energy of 120 mJ pulse-1 was reported. A more useful quantity for the reader would be the energy per pulse per square centimetre, as this more directly relates to absorption cross-sections (quoted in cm2 molecule) and consequent radical densities. Please supply this information / clarify. The same applies to the YAG laser energy (page 5 line 116). Technical: Page 3 line 63 – 65 was confusing. I think the authors

mean to say "Since all reaction paths for I, IO or OIO with H2O are endothermic" Page 3 line 86 "introduced in the reactor" should be something like "introduced to the reactor" The use of low-contrast colours on e.g. Fig 8 without other visual markers will make it very difficult for some readers to distinguish e.g. I2O2 from I2O3 from I2O4. Could dots or dashes be introduced to help with this issue of accessibility? Fig. 9 uses the same symbol type (circles) to represent I3O6, I3O7 and I5O12 – please make use of triangles / squares. Additionally, information was missing from the legend where only I5O12 is mentioned.

---

## Referee Comment (RC3) · Anonymous Referee #3 · 9 Jul 2020

The manuscript by Lewis et al. (acp-2020-456) describes experimental measurements and theoretical calculations to determine the absorption cross sections of higher iodine oxides, as well as modeling to assess the impact of the photochemistry in the atmosphere. The quality of the experimental data for what appear to be challenging experiments is reasonable. I have concerns that the modest theoretical methods applied may be inadequate to describe the electronically excited states in molecules such as the iodine oxides, but that is somewhat beyond my area of expertise. If the authors could demonstrate clearly that they are capable of reasonably predicting molecular properties of a well-known iodine oxide, it would go some way to assuaging those concerns. I am unable to comment on the atmospheric modeling, although I would note

[Figure]

that they rely on cross sections that have been determined experimentally at only two wavelengths. The paper is, for the most part, fairly well-written although the structure could be improved. Some of the text in the methods section would be better located in the results section. For example, lines 144-179 (including Figure 3 and 4) describe measurements, not the experimental set-up, and belong in the results section, as do the results of the ab initio calculations.

Table 1 compiles calculated geometries and vibrational frequencies, which are of little relevance to the subject matter of the paper and could be readily removed to supplementary information. On the other hand, no data for the calculated energies or oscillator strengths of the electronically excited states that are responsible for the visible and UV absorption are provided; the calculated ionization energy is also reported for only one species (IO3 on line 257). These data impact directly on the interpretation of the experimental results and should be compiled either in a revised version of Table 1 or in supplementary information.

The results section would benefit from a clearer introduction to describe what IxOy species are detected in the experiment and their time dependence. A figure showing the different "kinetic profiles", which are alluded to, would also be valuable. Presumably, the profiles have been characterized by varying the delay between the 248 nm photolysis pulse used to initiate the chemistry and the VUV photoionization pulse. The authors acknowledge (pages 13, 14) that fragmentation of larger species, can lead to signal increases at the masses of the photofragments, which would lead to possible systematic under-estimation of the depletion. To use the authors' example of I3O7, I have no sense of how much is present at the $\sim$7.6 ms time delay when the I2O4 (a potential daughter signal) is measured. Could other experimental parameters be varied (in principle) to modify the relative yields of different IxOy species to explore this in more detail? Relative photoionization cross sections for the various species will also play a role.

The depletion measurements in Figure 6 and 7, as well as the NO2 depletion used for

actinometry in Figure 4, are shown with an overlaid empirical fit. However, the fitting function is not described, or its choice explained, nor are any reasons for the different shapes discussed. Why does the effective "width" of the drop off appear to change for different species? Is it even meaningful? What delay does the time axis correspond to in these figures? The experiments use three laser pulses (two photolysis pulses, one to initiate the chemistry and a second to dissociate the iodine oxides) and one to detect the IxOy species. Clarity about which exactly which delay is being referred to would be helpful.

Minor comments.

Figure 1. The photolysis laser arguably should be labeled also with 532 nm as some experiments use that wavelength.

Line 113: Using the values for the cross section, concentration, and path length for I2 detection, I calculate OD = 0.82.

Figure 3. The caption could be more informative, for example, the time delay between the 248 nm pulse used to initiate the chemistry and the VUV photoionization pulse is not specified. Are the different colored traces the results of measurements at the optimum time for each?

---

## Referee Comment (RC4) · Anonymous Referee #4 · 10 Jul 2020

In this manuscript, Lewis et al. report experimental and theoretical determinations for the absorption cross-sections of iodine oxide species. Experimental measurements of the cross-sections were obtained using a technique that combined laser-flash photolysis (for production of iodine oxides from an $I_2/O_3$ mixture), photoionization mass spectrometry (for time-resolved species detection at 10.5 eV), and 355/532 nm laser photolysis. Cross-section measurements were calibrated using $NO_2$ and OIO at 355 nm and 532 nm, respectively. Theoretical determinations of the absorption spectra of the iodine oxides were obtained based on their B3LYP geometries. Atmospheric photolysis rates were determined across the actinic range using the theoretically-determined spectra, constrained by the experimental measurements, combined with the CAM-Chem model.

[Figure]

In the experimental and results sections, the authors describe using the kinetic profiles of the iodine oxide species to determine at what delay times the second photolysis laser (355/532 nm) was fired for the absorption cross-section measurements. It would therefore be appropriate to present the complete time profiles of each of the species (in the absence of the second photolysis laser pulse) either in the main text or the supplementary information.

Please give the form of the function that is used to fit the photodepletion data in Figures 4, 6 and 7. There appears to be significant differences in the depletion behavior of the different species that the authors do not address. For example, why is the depletion of $I_2O_4$ much sharper than $I_3O_7$?

What is the sensitivity of your measured cross-sections to the delay time of photolysis laser 2 (e.g. the 355 or 532 nm laser)?

What is the error in the theoretically determined absorption cross-sections? Does the magnitude of this error significantly impact the results of the atmospheric photolysis rates?

There is very limited discussion about the potential impact of the daughter ions (referred to in the manuscript as photofragments) of larger iodine oxides on the determination of the cross-sections of smaller iodine oxides. Given that not all of the iodine oxides produced in the experiment have the same cross-section, if larger iodine oxides undergo dissociative ionization at 10.5 eV to photofragments with the same exact mass as the smaller iodine oxides studied in this work, the measured depletions of the smaller iodine oxide species would be perturbed by the contribution of photofragments from the larger iodine oxide species. To what extent do the authors have evidence that this is not significantly hampering their cross-section measurements? Ionization energy calculations of larger iodine oxides to possible photofragments at the mass of the smaller iodine would indicate whether this is a concern or not at 10.5 eV. Additionally, measurements at reduced concentrations of [I], and at various delay times of the

second photolysis laser would provide further indications of potential interference.

Minor comments:

Figure 1 An entry port for H2O is indicated, is water used in any of these experiments?

Section 2.1 For clarity, it would be helpful to distinguish the three laser pulses used in these experiments using a numbering scheme (e.g. photolysis pulse 1 or 2, photoionization pulse).

Section 2.2 Table 1 and Figure 5 should be moved into the results section, or perhaps the supplementary information.

L239 The number of significant figures in the value and the error for the I5O12 cross-section are not consistent (and differ from the value in Table 2).

---

## Author Comment (AC1) · 27 Jul 2020

Paper Ref: acp-2020-456

Title: " Determination of the absorption cross-sections of higher order iodine oxides at 355 nm and 532 nm"

RESPONSE TO THE REPORT OF REVIEWER #1

We are grateful to the reviewer for helpful and constructive comments and suggestions. We address them point by point below. The Reviewer's comments are shown in bold typescript, our response in normal typescript. Changes to the manuscript are highlighted in red. Page numbers refer to the revised manuscript.

**Lewis et al combine experimental and theoretical work examining the absorption cross-sections of higher iodine oxides (IxOy). The results indicate that most oxides are rapidly removed by photolysis, which has repercussions for formation of iodine containing particle. The experiments are complex as IxOy have to be generated in-situ, the data appears to be of reasonable quality. This review does not address the accuracy of the theoretical work. The manuscript is well written; the authors may consider the following suggestions for improvement.**

**Main comment: Actinometry at 355 nm is using NO2 whereas OIO is used at 532 nm. The absorption cross-section of OIO at 532 nm is quoted as being "relatively well known" (L174). The cross-sections of I2 are well known as are those of NO2. It is questionable whether an unstable trace-gas, the spectrum of which has only been measured following its transient formation as the product of the IO-self reaction can be termed "well known". The authors must be more quantitative here. They should properly review (and cite) the limited literature on the OIO cross-section at this wavelength. The same applies to the unity quantum yield of photolysis of OIO. Do all the literature studies agree on this value? Again, the authors need to review (and cite) the literature and state why they believe the quantum yield is one, rather than simply citing one article in which one of the present authors also contributed.**

A problem with using $I_2$ as an actinometer is that its spectrum has many rovibrational lines around 532 nm, and therefore the cross sections determined at this wavelength are very resolution-dependent. Thus, we would have had to determine the absorption cross section of $I_2$ for our specific laser line width (1 cm$^{-1}$ or 0.03 nm). A second problem specific to our PI-ToF-MS detection system is that $I_2$ photoionizes very efficiently at 118 nm (see section 2.1), which causes an overload in the detector from the $I_2$ peak signal towards longer flight times (i.e. higher

masses) and requires gating the peak. We have nevertheless carried out a test with partial gating of the $I_2$ peak. In this test, $I_2$ depletion was $(17 \pm 5)\%$ and OIO depletion was $(54 \pm 4)\%$. Using the $I_2$ cross section at 532 nm determined by Tucceri et al. (2006) at 1.4 $cm^{-1}$ resolution (0.04 nm), we obtain $\sigma_{OIO} = (8 \pm 3) \times 10^{-18}$ $cm^2$ molecule$^{-1}$, which is fully consistent with the literature values of the absolute absorption cross sections of OIO listed in the table below, lending confidence to the method. The main source of uncertainty in this determination is the noise in the semi-gated $I_2$ trace.

| Reference | Resolution /nm | $\sigma_{OIO}$ /cm$^2$ molecule$^{-1}$ |
|---|---|---|
| Joseph et al. (2005) | 0.006 | 567.93 nm: $(1.51 \pm 0.18) \times 10^{-17}$
 532 nm*: $1.25 \times 10^{-17}$ |
| Gomez Martin et al. (2005)

 Spietz et al. (2005) | 0.35

 1.3 | 549.2 nm: $(1.3 \pm 0.3) \times 10^{-17}$

 549.2 nm: $(1.1 \pm 0.3) \times 10^{-17}$
 532 nm: $8.9 \times 10^{-18}$ |
| Tucceri et al. (2006) | 0.04 | 610 nm: $(6 \pm 2) \times 10^{-18}$
 532 nm*: $8.2 \times 10^{-18}$ |
| Bloss et al. (2001) | 1.13 | 549.2 nm: $(1.1 \pm 0.2) \times 10^{-17}$
 532 nm: $9.9 \times 10^{-18}$ |

The cross sections at 532 nm with an asterisk have been obtained by scaling the relative spectrum measured by Spietz et al. to the cross section of that study at the corresponding wavelength. The relative cross sections measured by Spietz et al. and Bloss et al. are in good agreement, as shown by the selected peak-to-valley ratios listed in the following table:

| Reference | $\sigma_{OIO}$(532.0 nm)/$\sigma_{OIO}$(541.2 nm) | $\sigma_{OIO}$(549.2 nm)/$\sigma_{OIO}$(560.3 nm) |
|---|---|---|
| Spietz et al. (2005) | $2.7 \pm 0.2$ | $3.2 \pm 0.2$ |
| Bloss et al. (2001) | $2.2 \pm 0.2$ | $3.0 \pm 0.2$ |

Considering the differences in methods, resolution and wavelength of the different determinations of the OIO cross sections listed above and the reported uncertainties, the agreement can be considered as reasonable and the cross section at 532 nm can be considered as well stablished, at least within 25%.

There are essentially two independent experimental determinations of the OIO photolysis quantum yield above 500 nm (Tucceri et al. 2006 and Gomez Martin et al. 2009). The studies by Tucceri et al. 2006 and Gomez Martin et al. 2009 employed similar methods and performed similar tests. The main problem of the reaction scheme used by Tucceri et al. in the detection of I atoms was the presence of high ozone concentrations ($\geq 10^{16}$ molecule cm$^{-3}$), which most likely reduced the lifetime of the I atom photofragment to tens of microseconds, besides other experimental complications such as formation of $I_2$ and aerosol. In the absence of $O_3$, Gomez Martin et al. observed a long-lived I photofragment and no reformation of OIO (on a time scale of several ms), strongly suggesting that OIO does indeed photolyze to $I + O_2$. Further evidence supporting a unit photolysis quantum yield comes from the short lifetime (200 fs) of the $C^2A_2$ excited state, determined by Ashworth et al. (2002) through simulations of the rotational envelopes of the observed absorption bands. A high-level MRCI theoretical study of the OIO excited states by Kirk Peterson (2010) found a low barrier for dissociation to $I + O_2$ for the $A^2B_2$ state, which should lead to efficient I atom production via an initial spin–orbit interaction between the $C^2A_2$ state and the nearby $B^2A_1$, followed by a strong vibronic interaction with the $A^2B_2$ state via an avoided crossing.

Insertion in Page 9, Line 225: The use of $I_2$ as actinometer is also precluded by the overload issue mentioned above. The OIO absorption cross section at 532 nm is relatively well known, within 25% of the average value of the four independent determinations reported in the literature (Bloss et al., 2001; Joseph et al., 2005; Spietz et al., 2005; Tucceri et al., 2006). On the other hand, conflicting results have been reported for the OIO photolysis quantum yield (Tucceri et al., 2006; Gomez Martin et al., 2009). Here we use the unit quantum yield reported by Gomez Martin et al. (2009), which was determined in a system free of interferences from ozone where a long-lived I atom photofragment and no reformation of OIO was observed over a time scale of several milliseconds. This result is also supported by the short lifetime (200 fs) of the excited state responsible for the observed absorption bands (Ashworth et al., 2002) and the existence of a feasible photolysis path revealed by high level ab initio calculations (Peterson, 2010).

**L31 Presumably the 9-16% ozone depletion from iodine chemistry refers to the marine boundary layer. This should be made clear.**

This % is indeed over the integrated tropical troposphere as indicated in the cited reference and a new reference added: https://www.atmos-chem-phys.net/16/1161/2016/

**L77 "This is currently an important uncertainty. . ." What does "this" refer to? Perhaps "an important source of uncertainty" is better?**

Amended in text

**L114 Signal-to-noise ratio ~ 400. S/N measured over what period of time? Is drift in laser intensity not more important than S/N for an absorption cell that has no reference photo-diodes. Limit of detection in OD units would be more useful.**

The signal from the photodiode was accumulated over ~15 seconds. It is true that the laser intensity drifts, however the drift was not significant over this time period. The S/N increases quickly over ~15 s, before plateauing, and eventually decreasing due to the drift in intensity. Signals were measured back-to-back, so as to negate intensity drift as much as possible. The back-to-back measurements were carried out until 3 concordant results (OD variation <10%) were obtained, which is typically the first 3 results.

Insertion in Page 5, Line 121: […] which corresponds to a minimum detectable OD of 2.5E-3. As there is no reference photodiode in this setup, drift in laser intensity must be accounted for. To negate the effect of laser drift as much as possible, the probe intensity is measured over a period of 15 seconds for both I and $I_0$, with ~10s between the measurements. To ensure that laser drift does not significantly affect the measured concentrations, measurements are taken until 3 concordant results are obtained (typically the first three results).

**Figure 4. What is the pulse-width of the excitation laser ? Please explain why the depletion of NO2 (presumably in ~10 ns) takes 150 ms until the new plateau is reached. To which expression was the red curve fitted and why was it chosen?**

The specified pulse width of the Continuum Surelite 10-ii is 3-5 ns. The depletion time of 150 **microseconds** is a sampling time and results from transport of molecules between the region where they are photolyzed to the region where they are photoionized. The sampling in this system can be described in the following steps:

    i)       Transport of molecules from the reactor toward the sampling orifice.

    ii)     Transport of the molecules through the sampling orifice.

    iii)    Transport of the molecules from the exit of the sampling orifice to the ionization region.

The sampling time depends on the alignment of the lasers and on the molecular masses of the bath gas and sample species (Baeza-Romero et al., 2011, DOI: 10.1002/kin.20620).

The curve chosen is a sigmoidal curve with variable Hill slope given by parameter 'p'.

$$y = S_0 + \frac{S_1 - S_0}{1 + 10^{(\log x_0 - x)p}}$$

This fitting function is chosen to fit empirically the observed depletion curves. The only parameters relevant to this study are the top and bottom asymptotes.

Insertion in Page 8, Line 169: The trace shown in Figure 4, which is similar to all the traces obtained in this experiment, exhibits a delay between the pre- and post-photodepleted signal. This delay corresponds to an instrumental sampling time depending on the alignment of the lasers and the molecular masses of the bath gas and the sample species (Baeza-Romero et al., 2012). In the present experiments, it was necessary to leave a small gap (~2 mm) between the photolysis volume and the sampling pinhole, so as to avoid hitting the skimmer cone with the laser. The diffusional exchange of molecules between the photodepleted volume, and the un-photodepleted volume immediately before the pinhole blurs the onset of the photodepletion as it is measured by the TOF-MS. The values of $S_0$ and $S_1$ were obtaining by empirically fitting the photodepletion trace to a sigmoidal function:

$$y = S_0 + \frac{S_1 - S_0}{1 + 10^{(\log x_0 - x)p}} \tag{2}$$

Fitting to this function ensures that the flat sections corresponding to the pre- and post-photodepletion concentrations ($S_0$ and $S_1$ respectively) are characterized in the precise regions outside of the aforementioned "blurred" zone. The parameters $x$ and $p$ are not of scientific importance for this study and are simply instrumental factors.

**L146 and L147 Laser "shot", excimer "flash". Perhaps use the term "pulse" ?**

Agreed that pulse is better, changed in the text

**Figure 6 What causes the increase in the IO signal after 1.85 ms.**

Photolysis of IO at 355 nm results in O and I atoms, which will react respectively with $I_2$ and $O_3$ in the reaction mixture rapidly, reforming some of the IO lost to photodepletion. In addition, some $O_3$ may also photolyze at 355 nm producing O atoms.

**L259 Please show the trace obtained using the actinometer (OIO). This could potentially go into supplementary information along with the raw data (lack of signal change) for the higher oxides and would give the reader an idea of the data-quality.**

Added as new Figure 8.

**Figure 7 Indicate when the laser was triggered in the I2O3 experiment. Are the red curves the same function as used in Figure 4 ?**

Now indicated in the figure caption.

Yes, all traces are analysed by performing the same empirical fit. This is now stated in page 8.

**L370 please provide references to the tropical UTLS being a "hotspot" of iodine chemistry. Have there been measurements of IO or higher iodine oxides in this part of the atmosphere?**

Yes, the reference was already there a few sentences later. We have inserted the reference (Koenig et al., PNAS 2020) at the beginning of the paragraph. However, calling the UTLS a 'hotspot' may lead to think of places like Mace Head, Antarctica, etc. Therefore, we refer now to the UTLS as an 'active iodine chemistry region' instead.

---

## Author Comment (AC2) · 27 Jul 2020

Paper Ref: acp-2020-456

Title: " Determination of the absorption cross-sections of higher order iodine oxides at 355 nm and 532 nm"

RESPONSE TO THE REPORT OF REVIEWER #2

We are grateful to the reviewer for helpful and constructive comments and suggestions. We address them point by point below. The Reviewer's comments are shown in bold typescript, our response in normal typescript. Changes to the manuscript are highlighted in red. Page numbers refer to the revised manuscript.

**Review of "Determination of the absorption cross-sections of higher order iodine oxides at 355 nm and 532 nm" by Lewis et al. This paper presents photochemical data for iodine oxides (IxOy), a hitherto elusive family of atmospheric transients thought to play an important role in the coastal marine boundary layer. Results were determined via laser photolysis / photodepletion experiments using mass-spectrometry detection of IxOy, supplemented by ab-initio calculations. This first extensive dataset can provide valuable information needed to construct models of atmospheric chemistry. However, there were a few points that require further clarification and discussion within the manuscript prior to publication in ACP.**

**Firstly, experiments appear to have been conducted under very different conditions of gas pressure and identity to those found in the boundary layer. At the top of page 6 it was stated that experimental pressures of 4 to 7 Torr were used (though whether of He or N2 was not clear). This is clearly a very different matrix of gases to 760 Torr of N2, O2 and H2O found in coastal boundary layer air. There is surely some doubt therefore, whether some qualitative observations from this work are valid for atmospheric models. The chemical mechanisms for IxOy formation will surely differ to some extent due to changes in stabilization rates for association products at the different gas-pressure, or perhaps reactions of transients with O2 (or even H2O) that would proceed faster in the boundary layer. Specifically, much is made of the absence of I4Oy species, but we simply do not know if these compounds would be formed in realistic atmospheric conditions. If there is clear evidence for why such pressure or O2 effects are unimportant then this needs to be detailed in the manuscript.**

The carrier used was He in all photolysis experiments. We report results elsewhere (Gomez Martin et al, 2020) showing that iodine oxide clusters with even number of iodine atoms do not form, and that $I_xO_y + H_2O$ reactions are extremely slow, either at low pressure or at higher pressures and with a more atmospherically representative matrix of gases. We have added this reference since this manuscript has very recently been accepted for publication in *Nat. Comm.*

Insertion in Page 12, Line 335:  […], even for higher pressures and using $N_2$ as carrier gas (Gómez Martín et al., 2020)

**Second, regarding more quantitative results, could photolysis quantum yields and therefore photolysis cross-sections differ as the pressure and identity of the surrounding gas matrix changes? Certainly these effects can be important for many atmospheric transients, not least for the actinometer NO2 though at a longer wavelength than used in this work. I suspect that such pressure effects will have a negligible impact on the quantitative results from this work, given, as stated on page 16, the featureless nature of the spectra. However, given the large divergence from atmospheric conditions noted above, a strong statement to the effect that these cross-sections / quantum yields are applicable to realistic atmospheric conditions would be advisable.**

As the reviewer points out, $NO_2$ excitation at 355 nm is not dependent on the nature or pressure of the matrix, and is therefore suitable for use as an actinometer in these experiments (Troe, J. *Z. Phys. Chem.* 2000, **214**, p. 573-581). It is important to separate the two points being made here: firstly, the potential effect of the matrix on the chemistry leading to the formation of the molecules being studied in this work and secondly, the potential effect of the matrix on the photolysis quantum yield of the molecules being photolysed. Assuming the molecules are thermally equilibrated (a safe assumption at the pressures and timescales used in this study), the route to their formation has no bearing on the photolytic properties of the molecules. As the reviewer points out, the broad, featureless calculated absorption spectra indicate excitation to an unbound upper state. The photolysis processes in this study therefore do not depend on the nature and pressure of the matrix, and are suitable for use in atmospheric models.

Insertion in Page 9, Line 236:  This is a reasonable assumption for broad band absorption spectra indicating excitation to an unbound upper state. The photolysis processes in this study therefore do not depend on the nature and pressure of the carrier gas matrix, and the results can be applied directly in atmospheric models.

**Third regards the use of OIO as the actinometer for 532 nm experiments. I can understand why this molecule was used, given a limited set of alternatives. However, the manuscript needs to properly represent the problems that this introduces into the interpretation of results. The cross-section for OIO may be "relatively well known" as stated on page 8 (line 176), but I was not able to find a recent review justifying this statement. On the contrary, of the two references quoted in this work for OIO cross sections, only one (from Bloss et al.) quotes a value at 532 nm. A very quick look in the literature yielded cross-section data from five additional papers (Spietz 2005, Joseph 2005, Himmelmann 1996, Tucceri 2006 and from Ingham 2000) that differ by up to a factor of two at some wavelengths. Further, the quantum yield (QY) for atomic iodine production from OIO is certainly not established. Ingham et al. reported QY < 0.15 for this channel, in direct contrast to the value of unity taken from Gomez Martin et al. and used for the purposes of this work. Happily, a careful re-wording of the manuscript here can rectify these problems. First, the section on page 8 needs to take full account of the published literature on OIO. Second, the results obtained in this work need to be clearly denoted as being determined relative to the rather uncertain cross-sections and quantum yields for OIO at 532 nm.**

Reviewer #1 has raised the same question. We have included in the revised manuscript a review of both the absorption cross section and the photolysis quantum yield of OIO at 532 nm. Note that Gomez Martin 2005 and Spietz et al. 2005 are part of the same body of work. While the former paper reported the cross section at a single wavelength and a higher resolution, the later reported the visible spectrum at a lower resolution. Overall, the four existing independent determinations do not deviate by more than ~25% from the average when they are extrapolated to 532 nm, and the relative cross sections measured by Bloss et al. and Spietz et al. across the visible spectrum are in very good agreement.

Insertion in Page 9, Line 225: The OIO absorption cross section at 532 nm is relatively well known, within 25% of the average value of the four independent determinations reported in the literature (Bloss et al., 2001; Joseph et al., 2005; Spietz et al., 2005; Tucceri et al., 2006). On the other hand, conflicting results have been reported for the OIO photolysis quantum yield (Tucceri et al., 2006; Gómez Martín et al., 2009). Here we use the unit quantum yield reported by Gomez Martin et al. (2009), which was determined in a system free of interferences from ozone where a long-lived I atom photofragment and no reformation of OIO was observed over a time scale of several milliseconds. This result is also supported by the short lifetime (200 fs)

of the excited state responsible for the observed absorption bands (Ashworth et al., 2002) and the existence of a feasible photolysis path revealed by high level ab initio calculations (Peterson, 2010).

**Lastly, on Figure 4 "an empirical fit" was used to obtain depletion parameters. What was the function? More interestingly, which processes were responsible for the delay in signal depletion following (presumably rapid) photolysis. Is this delay instrumental, and therefore present in other experiments? It is not possible for the reader to assess for themselves, as a very different timescale is presented on Figs 6 and 7.**

This point has also been raised by Reviewer #1 and we have included a thorough explanation and a reference (Baeza-Romero et al., 2012) in the revised version of the manuscript.

**More minor comments:**

**Page 2 it was stated that the main atmospheric fate of iodine atoms is reaction with O3 to form IO. This is likely true across much of the globe, but a significant alternative exists in polluted air (as encountered in many important areas of the coastal MBL) where reaction with NO2 to form INO2 would be competitive.** In an iodine-rich, semi-polluted environment like Roscoff (France), the $NO_2$ mixing ratios range between a few hundred ppt to a few ppb and the $O_3$ mixing ratios range from 10 to 50 ppb (Mahajan et al, 2009). Assuming the average mixing ratios registered in Roscoff during the RHaMBLe campaign, i.e. ~1 ppb $NO_2$ and ~30 ppb $O_3$, the ratio of first-order loss rates of atomic iodine to reaction with $O_3$ and $NO_2$ is ~7. Thus, it is a fair statement to say that the main atmospheric fate of atomic iodine is to react with ozone, even in semi-polluted, coastal environments.

**The text on page 3 reports flows diluted in He whilst Figure 1 appears to indicate N2 as the principal diluent. Which is correct? If a mixture of the two then please use the text and / or the caption to Figure 1 to offer more detail.**

Helium is correct. We have changed Figure 1 accordingly.

**Similarly in the experimental details a laser energy of 120 mJ pulse-1 was reported. A more useful quantity for the reader would be the energy per pulse per square centimetre, as this more directly relates to absorption cross-sections (quoted in cm2 molecule) and consequent radical densities. Please supply this information / clarify.**

Clarified in text to give the total laser energy per pulse and the laser energy per pulse per unit area (i.e. the laser pulse fluence).

**The same applies to the YAG laser energy (page 5 line 116).**

Clarified to give laser fluence (energy per pulse per unit area).

**Technical: Page 3 line 63 – 65 was confusing.**

Clarified. The CI-ToF-MS technique referred to uses a $NO_3^-$ ion source. The observation of $IO_3^-$ signals is interpreted as a result of ambient $HIO_3$ being ionized by $NO_3^-$.

**I think the authors mean to say "Since all reaction paths for I, IO or OIO with H2O are endothermic"**

Changed.

**Page 3 line 86 "introduced in the reactor" should be something like "introduced to the reactor"**

Amended in text

**The use of low-contrast colours on e.g. Fig 8 without other visual markers will make it very difficult for some readers to distinguish e.g. I2O2 from I2O3 from I2O4. Could dots or dashes be introduced to help with this issue of accessibility?**

Done  (now Figure 9)

**Fig. 9 uses the same symbol type (circles) to represent I3O6, I3O7 and I5O12 – please make use of triangles / squares. Additionally, information was missing from the legend where only I5O12 is mentioned**

Done  (now Figure 10)

---

## Author Comment (AC3) · 27 Jul 2020

Paper Ref: acp-2020-456

Title: " Determination of the absorption cross-sections of higher order iodine oxides at 355 nm and 532 nm"

RESPONSE TO THE REPORT OF REVIEWER #3

We are grateful to the reviewer for helpful and constructive comments and suggestions. We address them point by point below. The Reviewer's comments are shown in bold typescript, our response in normal typescript. Changes to the manuscript are highlighted in red. Page numbers refer to the revised manuscript.

**The manuscript by Lewis et al. (acp-2020-456) describes experimental measurements and theoretical calculations to determine the absorption cross sections of higher iodine oxides, as well as modeling to assess the impact of the photochemistry in the atmosphere. The quality of the experimental data for what appear to be challenging experiments is reasonable. I have concerns that the modest theoretical methods applied may be inadequate to describe the electronically excited states in molecules such as the iodine oxides, but that is somewhat beyond my area of expertise. If the authors could demonstrate clearly that they are capable of reasonably predicting molecular properties of a well-known iodine oxide, it would go some way to assuaging those concerns.**

Following the suggestion of the reviewer, we have compared the experimental absorption spectra of IO, OIO and HOI with the corresponding spectra calculated with the TD-DFT method. This comparison is shown in Figure A2 (below). It can be seen that even for radicals like IO and OIO, the method gives a reasonable prediction of the spectral position of the electronic bands, and average absorption cross sections of the same order. For closed-shell molecules, the agreement is expected to be better, as demonstrated for HOI.

We have introduced the following sentence in Page 10, at the end of section 2.2: Although there are more advanced methods for the calculation of electronic spectra, TD-DFT offers a reasonable compromise between low computational cost and accuracy of the predicted transitions. Figure A2 shows a comparison between the experimental and TD-DFT absorption spectra of IO, OIO and HOI. Note that although the TD-DFT method is not designed to predict ro-vibrational structure, the spectral positions of the electronic bands and the average absorption cross sections are in reasonable agreement with the experiment, even in the case of

open shell species like IO and OIO. Higher iodine oxides are closed shell molecules and the accuracy of the transitions is expected to be similar to the result for HOI.

The new figure included in the Appendix A (Figure A2) is the following:

[Figure]

Figure A2: Experimental absorption cross sections of IO (Sander et al., 2011), OIO (Spietz et al., 2005) and HOI (Sander et al., 2011) and absorption spectra calculated with the TD-DFT method (this work).

**I am unable to comment on the atmospheric modeling, although I would note that they rely on cross sections that have been determined experimentally at only two wavelengths.**

Even though the experiments are limited to two wavelengths, this is significant progress with respect to what was known before. We use the two single-wavelength experimental cross sections to show that spectra measured by absorption spectroscopy cannot be directly used in

the determination of photolysis rates and to validate calculated spectra, which do not suffer from absorption overlap issues. Of course, future studies should address experimentally the wavelength dependence of the absorption spectra of iodine oxides, e.g. by using the same approach presented in our work but with a tunable laser instead of a Nd:YAG laser.

**The paper is, for the most part, fairly well-written although the structure could be improved. Some of the text in the methods section would be better located in the results section. For example, lines 144-179 (including Figure 3 and 4) describe measurements, not the experimental set-up, and belong in the results section, as do the results of the ab initio calculations.**

This paper is not concerned with demonstrating the observation of $I_xO_y$ by PI-ToF-MS, which has been already reported in previous publications. Since this method of detection of $I_xO_y$ is already proven, we do not think that Figure 3 belongs to the results section, but rather to the methods section as a proof that we can see these species.

We have introduced the following sentence in Page 6, Line 137: Successful detection of $I_xO_y$ by this method has been demonstrated elsewhere (Gomez Martin et al., 2013; Wei et al., 2017; Gomez Martin et al., 2020).

Similarly, Figure 4 and the corresponding discussion also deal with methodological considerations rather than with new results. The use of PI-TOF-MS in photochemical experiments has been discussed previously (Baeza-Romero et al, 2012). New text has been inserted in p. 8 after Figure 4 in response to another concern of the reviewers (see below), which further helps to appreciate the methodological nature of this section of the paper.

Regarding the ab initio calculations, we consider the determination of the ground state geometries and molecular parameters of iodine oxides as merely methodological, since there are previous publications where these have been reported at different levels of theory.

Insertion in p. 10, L245: Note that the ground states some of these oxides have been studied at a higher level of theory elsewhere (Kaltsoyannis and Plane, 2008; Galvez et al., 2013)

The novelty of these ab initio calculations is the determination of absorption spectra. We have added a new subsection 3.3 where the ab initio spectra are formally mentioned as results, and additional information about the spectra is provided.

Addition in Page 16:

**3.3. Ab initio spectra**

The calculated spectra are displayed in Figure 9 ($I_2O_2$, $I_2O_3$ and $I_2O_4$), Figure 10 ($I_3O_6$ and $I_3O_7$), and Figure 11 ($IO_3$). Oscillator strengths of the electronic transitions that are responsible for the visible and UV absorptions are provided in Appendix A. The TD-DFT spectra were wavelength-shifted by applying a constant energy shift to get agreement with the experiment at 355 nm. The shifts are quite modest, within the expected error at this level of theory (Foreman and Frisch 2015): $I_2O_3$ (30 kJ mol$^{-1}$), $I_2O_4$ (-12 kJ mol$^{-1}$), $I_3O_6$ (9.2 kJ mol$^{-1}$), $I_3O_7$ (-21 kJ mol$^{-1}$). Applying a constant energy shift means assuming that all the excited state energies are offset by a constant amount with respect to the ground state.

**Table 1 compiles calculated geometries and vibrational frequencies, which are of little relevance to the subject matter of the paper and could be readily removed to supplementary information. On the other hand, no data for the calculated energies or oscillator strengths of the electronically excited states that are responsible for the visible and UV absorption are provided; the calculated ionization energy is also reported for only one species (IO3 on line 257). These data impact directly on the interpretation of the experimental results and should be compiled either in a revised version of Table 1 or in supplementary information.**

Table 1 and Figure 5 have been removed from the main text and are now included in Appendix A following the reviewer request. The calculated oscillator strengths have been included in Appendix A as well.

Calculated and experimental ionization energies of $I_xO_y$ species relevant to this work (except $IO_3$) have been reported previously, and the corresponding papers (Gómez Martín et al., 2013; Wei et al., 2017; Gómez Martín et al., 2020) are properly cited in the paper (page 6).

**The results section would benefit from a clearer introduction to describe what IxOy species are detected in the experiment and their time dependence. A figure showing the different "kinetic profiles", which are alluded to, would also be valuable. Presumably, the profiles have been characterized by varying the delay between the 248 nm photolysis pulse used to initiate the chemistry and the VUV photoionization pulse.**

Why have clarified the species detected and the meaning of the "kinetic profiles" in the experimental section. Another wording for the same concept is "time trace". The reviewer is

correct in that time is defined as the delay between excimer photolysis and VUV photoionization.

Changes in Page 7, Lines 175-160: Each experiment results in a 3-dimensional dataset of signal intensity (proportional to concentration) vs. 248 nm photolysis - VUV photoionization delay time (kinetic profile or time trace) and time-of-flight (mass spectrum). Figure 3 shows mass spectra with the most prominent peaks obtained at different delay times . Mass-to-charge (m/z) calibration of time-of-flight was performed by selecting a number of well-known prominent mass peaks (e.g. IO at m/z =143, OIO at m/z =159, I2O3 at m/z =302, I2O4 at m/z =318 and I3O7 at m/z =493 (Gómez Martín et al., 2013)).

We now refer specifically to Figure 3 in the results section to make clearer what are the target species in the context of the mass spectrum. We have added a better introduction to the Results section, moving here some material from the Methods section describing how fragmentation problem affects our results and adding the requested figure with kinetic profiles:

As described in the experimental section, kinetic profiles of the growth and removal of the target iodine oxide species shown in Figure 3 were carried out in order to define the time periods with the most suitable kinetic profiles for photolysis measurements (Figure 5). Fragmentation of iodine oxides was a significant problem in these experiments, as predicted from ionization energy calculations of larger iodine oxides to possible photofragments at 10.5 eV (Gomez Martin et al., 2020). High amounts of active iodine ($IO_x$ = I, IO) released from reaction R12 lead to fast formation of $I_xO_y$ and particles. Under these conditions, at long times after the peak IO and OIO (~3-5 ms), the observed signal of IO, OIO and $I_xO_y$ is contaminated by photofragmentation of higher order iodine oxides. For this reason, great care needed to be taken to establish a time window for each species wherein higher oxides are not present, to ensure that any depletion in the mass spectrometric signal for each species is exclusively due to the removal of the species via photolysis (Figure B1). Evidence of fragmentation comes in the form of a secondary growth in the signal seen for IO and OIO (Figure 5). The delay between the excimer and the Nd:YAG photolysis laser was therefore carefully selected to coincide with a period of relatively constant signal of the desired analyte, typically a maximum for short lived species, or a slow rise for larger reaction products of interest.

[Figure]

**Figure 5: Time traces of IO, OIO, $I_2O_3$, $I_2O_4$, $I_3O_7$ and $I_5O_{12}$ from -1-10 ms, at 1 ms intervals for a mixture of He (10 torr), $O_3$ and $I_2$ ([$O_3$] = 4 × 10$^{14}$ molecule cm$^{-2}$ [$I_2$] = 2.8 × 10$^{14}$ molecule cm$^{-2}$) flash photolyzed by an excimer pulse at t = 0 (130 mJ pulse$^{-1}$). The red sections highlighted for each species correspond to the optimal delay windows for photolysis of the corresponding species for this set of conditions.**

Kinetic profiles of IO, OIO, $I_2O_3$, $I_2O_4$, $I_3O_7$ and $I_5O_{12}$ are shown in Figure 5, along with 0.5 ms windows within which each species can be photolysed free of contamination from daughter ions of larger $I_xO_y$ species. To further illustrate the lack of contamination from higher oxides, the mass spectra collected during the windows outlined in the kinetic traces are averaged, showing the target species to be present absent of higher oxides (new Figure B1). The reviewer is correct to assume that contribution to a signal of interest from daughter ions of larger $I_xO_y$ species would cause erroneous measurement of photodepletion in the species of interest, and as such, great care was taken to eliminate the possibility of the aforementioned contamination.

Note that the kinetic traces were obtained at the beginning of each experimental session, for each set of conditions, in order to elucidate the optimal time delay for a species of interest, whereby the signal was present, but not larger species, the daughter ions of which would contaminate the desired signal. The kinetic trace was generally not averaged for a long period of time, since good signal to noise is not required, and is just a preliminary check used to establish the correct timings for the photolysis experiments. This discussion appears now in Appendix B:

**APENDIX B. Photofragmentation of $I_xO_y$ species**

The photofragmentation of $I_xO_y$ species to daughter ions in the photoionization chamber of the detection apparatus necessitates careful experimentation in order to ensure that any photodepletion of a species of interest is solely due to its 355 nm or 532 nm photolysis in the flow tube, and not obscured by the daughter ions of larger $I_xO_y$ species. To elucidate the optimal window for investigating photodepletion of each species, time resolved mass-spectra were recorded for each set of experimental conditions (Figure 5). From the kinetic information, it is then possible to inspect the averages of the mass spectra within these windows to ensure that the species of interest is present, but larger $I_xO_y$ species are not, as shown in Figure B1. By ensuring no larger $I_xO_y$ species are present, it follows that for a species of interest, only parent ions of the species are contributing to the signal intensity within that time window, and that no contribution to the recorded signal is coming from daughter ions. Note that the signals shown in Figure 5 and Figure B1 are not accumulated for extended periods of time, and as such are relatively noisy. Long accumulation times and corresponding large signal to noise ratios are unnecessary for these experiments, since the objective is simply to elucidate optimal time-delays within which photodepletion experiments are carried out, (photodepletion experiments are typically carried for ~10× as many accumulations) and are carried out prior to an experimental session. It should be noted also that the optimal timescales such as those shown in Figure 5 and Figure B1, vary depending on the concentration of IO formed at the beginning of the reaction sequence, and since the reactions which facilitate the stepwise formation of the higher oxides are second-order, even modest changes in [IO] at early times can result in significant changes to the appearance times of the different species of interest.

[Figure]

**Figure B1: Mass spectra corresponding to the optimised time delays shown in Figure 5. The plots are generated by averaging the signal obtained for each mass over the 5 ms window. The species of interest is given in the top right corner of each spectrum.**

The authors acknowledge (pages 13, 14) that fragmentation of larger species, can lead to signal increases at the masses of the photofragments, which would lead to possible systematic under-estimation of the depletion. To use the authors' example of I3O7, I have no sense of how much is present at the _7.6 ms time delay when the I2O4 (a potential daughter signal) is measured. Could other experimental parameters be varied (in

**principle) to modify the relative yields of different IxOy species to explore this in more detail? Relative photoionization cross sections for the various species will also play a role.**

This question is answered in conjunction with the previous question in the above response, and has been clarified with the aid of new figures in the main text and Appendix B.

**The depletion measurements in Figure 6 and 7, as well as the NO2 depletion used for actinometry in Figure 4, are shown with an overlaid empirical fit. However, the fitting function is not described, or its choice explained, nor are any reasons for the different shapes discussed. Why does the effective "width" of the drop off appear to change for different species? Is it even meaningful? What delay does the time axis correspond to in these figures? The experiments use three laser pulses (two photolysis pulses, one to initiate the chemistry and a second to dissociate the iodine oxides) and one to detect the IxOy species. Clarity about which exactly which delay is being referred to would be helpful.**

These points have been raised by Reviewers #1 and #2 as well. A detailed answer can be found in the response to Reviewer #1. We added the following piece of text to the manuscript to answer the reviewers' concerns:

Insertion in Page 8, Line 196: The trace shown in Figure 4, which is similar to all the traces obtained in this experiment, exhibits a delay between the pre- and post-photodepleted signal. This delay corresponds to an instrumental sampling time depending on the alignment of the lasers and the molecular masses of the bath gas and the sample species (Baeza-Romero et al., 2012). In the present experiments, it was necessary to leave a small gap (~2 mm) between the photolysis volume and the sampling pinhole, so as to avoid hitting the skimmer cone with the laser. The diffusional exchange of molecules between the photodepleted volume, and the un-photodepleted volume immediately before the pinhole blurs the onset of the photodepletion as it is measured by the TOF-MS. The values of $S_0$ and $S_1$ were obtaining by empirically fitting the photodepletion trace to a sigmoidal function:

$$y = S_0 + \frac{S_1 - S_0}{1 + 10^{(\log x_0 - x)p}} \tag{2}$$

Fitting to this function ensures that the flat sections corresponding to the pre- and post-photodepletion concentrations ($S_0$ and $S_1$ respectively) are characterized in the precise regions outside of the aforementioned "blurred" zone. The parameters $x$ and $p$ are not of scientific importance for this study and are simply instrumental factors.

**Minor comments.**

**Figure 1. The photolysis laser arguably should be labeled also with 532 nm as some experiments use that wavelength.**

Done

**Line 113: Using the values for the cross section, concentration, and path length for I2 detection, I calculate OD = 0.82.**

Apologies, 0.82 is correct, we mistakenly used the path length as 30 cm (the length of the Herriott cell) instead of 40 cm (the length of the single pass cell). This error is only in the specific example in the paper and the correct value of 40 cm was used in the work itself.

**Figure 3. The caption could be more informative, for example, the time delay between the 248 nm pulse used to initiate the chemistry and the VUV photoionization pulse is not specified. Are the different colored traces the results of measurements at the optimum time for each?**

The figure shows a complete time trace, averaged along the time axis, and is only intended to show a clear picture of the species being studied in this experiment, to give the reader an idea of how the peaks appear in the mass spectrum. The caption has been updated to clarify this fact.

---

## Author Comment (AC4) · 27 Jul 2020

Paper Ref: acp-2020-456

Title: " Determination of the absorption cross-sections of higher order iodine oxides at 355 nm and 532 nm"

RESPONSE TO THE REPORT OF REVIEWER #4

We are grateful to the reviewer for helpful and constructive comments and suggestions. We address them point by point below. The Reviewer's comments are shown in bold typescript, our response in normal typescript. Changes to the manuscript are highlighted in red. Page numbers refer to the revised manuscript.

**In this manuscript, Lewis et al. report experimental and theoretical determinations for the absorption cross-sections of iodine oxide species. Experimental measurements of the cross-sections were obtained using a technique that combined laser-flash photolysis (for production of iodine oxides from an I2/O3 mixture), photoionization mass spectrometry (for time-resolved species detection at 10.5 eV), and 355/532 nm laser photolysis. Cross-section measurements were calibrated using NO2 and OIO at 355 nm and 532 nm, respectively.**

**Theoretical determinations of the absorption spectra of the iodine oxides were obtained based on their B3LYP geometries. Atmospheric photolysis rates were determined across the actinic range using the theoretically-determined spectra, constrained by the experimental measurements, combined with the CAM-Chem model. In the experimental and results sections, the authors describe using the kinetic profiles of the iodine oxide species to determine at what delay times the second photolysis laser (355/532 nm) was fired for the absorption cross-section measurements. It would therefore be appropriate to present the complete time profiles of each of the species (in the absence of the second photolysis laser pulse) either in the main text or the supplementary information.**

The same point has been raised by Reviewer #3. New Figure 5 includes some examples of full time traces as requested (the Figure is included in the response to Reviewer #3)

**Please give the form of the function that is used to fit the photodepletion data in Figures 4, 6 and 7. There appears to be significant differences in the depletion behavior of the different species that the authors do not address. For example, why is the depletion of I2O4 much sharper than I3O7?**

This point has been raised by Reviewers #1, #2 and #3, and has been clarified in the corresponding responses

**What is the sensitivity of your measured cross-sections to the delay time of photolysis laser 2 (e.g. the 355 or 532 nm laser)?**

The cross sections are insensitive to the delay time of the second photolysis laser, provided the species of interest is present, and uncontaminated by daughter ions of higher oxides. Great care was taken to assure that this was the case for all species being measured (see new Appendix B below).

**What is the error in the theoretically determined absorption cross-sections? Does the magnitude of this error significantly impact the results of the atmospheric photolysis rates?**

One thing we should have mentioned is that the TD-DFT spectra were wavelength shifted by applying a constant energy shift to get agreement with the experiment at 355 nm. The shifts are quite modest, within the expected error at this level of theory (Foreman and Frisch, 2015): $I_2O_3$ (30 kJ mol-1), $I_2O_4$ (-12 kJ mol-1), $I_3O_6$ (9.2 kJ mol$^{-1}$), $I_3O_7$ (-21 kJ mol$^{-1}$). Applying a constant energy shift means assuming that all the excited state energies are offset by a constant amount with respect to the ground state. As shown in Figure 10, the upper limits from experiment are within 20% of the theoretical calculations for linear $I_3O_6$ and $I_3O_7$.

Having anchored the theoretical absorption spectrum at 355 nm, we then assume that photolysis is possible up to the dissociation limit (which is the case for all the transitions of the $I_xO_y$ (x > 1) molecules). A conservative estimate is a factor of 2 uncertainty in the *J* values for these molecules, which does not change the conclusions of our study.

We have introduced the following changes in the manuscript to clarify these points:

Page 16, first paragraph: The TD-DFT spectra were wavelength-shifted by applying a constant energy shift to get agreement with the experiment at 355 nm. The shifts are quite modest, within the expected error at this level of theory (Foreman and Frisch, 2015): $I_2O_3$ (30 kJ mol$^{-1}$), $I_2O_4$ (-12 kJ mol$^{-1}$), $I_3O_6$ (9.2 kJ mol$^{-1}$), $I_3O_7$ (-21 kJ mol$^{-1}$). Applying a constant energy shift means assuming that all the excited state energies are offset by a constant amount with respect to the ground state.

Page 17, Lines 407-409: The calculated spectra shown in Figure 9 and Figure 10 agree generally well with the experimentally determined values at 355 nm and 532 nm after small

wavelength shifts indicated above. As shown in Figure 10, the upper limits from experiment are within 20% of the theoretical calculations for linear $I_3O_6$ and $I_3O_7$.

Page 20, Lines 463-465: Having anchored the theoretical absorption spectra at 355 nm, we then assume that photolysis is possible up to the dissociation limit (which is the case for all the transitions of the $I_xO_y$ (x > 1) molecules). A conservative estimate is a factor of 2 uncertainty in the $J$ values for these molecules.

**There is very limited discussion about the potential impact of the daughter ions (referred to in the manuscript as photofragments) of larger iodine oxides on the determination of the cross-sections of smaller iodine oxides. Given that not all of the iodine oxides produced in the experiment have the same cross-section, if larger iodine oxides undergo dissociative ionization at 10.5 eV to photofragments with the same exact mass as the smaller iodine oxides studied in this work, the measured depletions of the smaller iodine oxide species would be perturbed by the contribution of photofragments from the larger iodine oxide species. To what extent do the authors have evidence that this is not significantly hampering their cross-section measurements?**

The discussion about the impact of fragmentation has been expanded, including an Appendix B. Additional plots have been generated to illustrate the steps taken to eliminate this important potential effect highlighted by the reviewer (Figure B1). Essentially, kinetic profiles were taken for each set of conditions (new Figure 5), which allowed the optimal window for photolysis to be established, whereby the signal for a species is present in the spectrum, but any larger species which could photofragment to a daughter ion indistinguishable from the species of interest, is not present. In this way, it can be said with confidence, that daughter-ion contamination is not an issue in these experiments.

Addition to the manuscript:

**APENDIX B. Photofragmentation of $I_xO_y$ species**

The photofragmentation of $I_xO_y$ species to daughter ions in the photoionization chamber of the detection apparatus necessitates careful experimentation in order to ensure that any photodepletion of a species of interest is solely due to its 355 nm or 532 nm photolysis in the flow tube, and not obscured by the daughter ions of larger $I_xO_y$ species. To elucidate the optimal window for investigating photodepletion of each species, time resolved mass-spectra were recorded for each set of experimental conditions (Figure 5). From the kinetic information, it is then possible to inspect the averages of the mass spectra within these windows to ensure that

the species of interest is present, but larger $I_xO_y$ species are not, as shown in Figure B1. By ensuring no larger $I_xO_y$ species are present, it follows that for a species of interest, only parent ions of the species are contributing to the signal intensity within that time window, and that no contribution to the recorded signal is coming from daughter ions. Note that the signals shown in Figure 5 and Figure B1 are not accumulated for extended periods of time, and as such are relatively noisy. Long accumulation times and corresponding large signal to noise ratios are unnecessary for these experiments, since the objective is simply to elucidate optimal time-delays within which photodepletion experiments are carried out, (photodepletion experiments are typically carried for ~10× as many accumulations) and are carried out prior to an experimental session. It should be noted also that the optimal timescales such as those shown in Figure 5 and Figure B1, vary depending on the concentration of IO formed at the beginning of the reaction sequence, and since the reactions which facilitate the stepwise formation of the higher oxides are second-order, even modest changes in [IO] at early times can result in significant changes to the appearance times of the different species of interest.

[Figure]

**Figure B1: Mass spectra corresponding to the optimised time delays shown in Figure 5. The plots are generated by averaging the signal obtained for each mass over the 5 ms window. The species of interest is given in the top right corner of each spectrum.**

**Ionization energy calculations of larger iodine oxides to possible photofragments at the mass of the smaller iodine would indicate whether this is a concern or not at 10.5 eV.**

Fragmentation is indeed a concern, as demonstrated by the traces of IO and OIO at longer delay times. Ab initio calculations of ionization and fragmentation energies are included in a recently accepted paper (Gomez Martin et al. 2020). We have included a reference in the paragraph introducing the results.

**Additionally, measurements at reduced concentrations of [I], and at various delay times of the second photolysis laser would provide further indications of potential interference.**

Appendix B explains the procedure for minimising the influence of daughter ions.

**Minor comments: Figure 1 An entry port for H2O is indicated, is water used in any of these experiments?**

Water was not used in these experiments. The Figure has been updated.

**Section 2.1 For clarity, it would be helpful to distinguish the three laser pulses used in these experiments using a numbering scheme (e.g. photolysis pulse 1 or 2, photoionization pulse).**

For clarity, the laser pulses have been amended in the text, and prefaced with either excimer, photolysis or PI laser.

**Section 2.2 Table 1 and Figure 5 should be moved into the results section, or perhaps the supplementary information.**

A new Appendix A has been added containing supplementary ab initio results.

**L239 The number of significant figures in the value and the error for the I5O12 cross section are not consistent (and differ from the value in Table 2).**

Amended in text

---

## Author Response (AR2)

Paper Ref: acp-2020-456

Title: " Determination of the absorption cross-sections of higher order iodine oxides at 355 nm and 532 nm"

RESPONSE TO THE EDITOR'S TECHNNICAL COMMENTS

We are grateful to the editor for helpful corrections. We address them point by point below. The Editor's comments are shown in bold typescript, our response in normal typescript.

**Comments to the Author:**

**L30: remove space before period**

Done

**L112: insert space**

Done

**L114: , giving -> gives**

**Figure 3 and Figure B1: In principle, m/z needs no units in the X-axis title (e.g., see the discussion in https://www.degruyter.com/view/journals/pac/85/7/article-p1515.xml)**

Done

**L274: atomic mass units, which is the value of m/z for I2O5 -> m/z units corresponding to I2O5**

Done

**L283 and in other places in the text: parent ion -> precursor ion; daughter ion -> product ion (e.g., see https://www.degruyter.com/view/journals/pac/85/7/article-p1515.xml)**

Done

**L302: remove space before period**

Done

**L403: Figure 13(d-f) show -> Figures 13(d-f) show**

Done

**L500: missing volume and pages**

Done

**L538: any updates to this reference?**

By editorial request the tittle has been changed. A DOI has not been assigned yet.

**L584: alpha replaced by a special character**

Done

**L631, 633, 636, 649, and few other places: sub/superscripts in reference titles**

Done